# Bulk-Boundary Correspondence and Exceptional Points for a Dimerized Hatano-Nelson Model with Staggered Potentials

Yasamin Mardani,[1, *] Rodrigo A. Pimenta,[1, †] and Jesko Sirker[1, ‡]

[1]*Department of Physics and Astronomy, University of Manitoba, Winnipeg R3T 2N2, Canada*
(Dated: October 3, 2024)

It is well-known that the standard bulk-boundary correspondence does not hold for non-Hermitian systems in which also new phenomena such as exceptional points do occur. Here we study by analytical and numerical means a paradigmatic one-dimensional non-Hermitian model with dimerization, asymmetric hopping, and imaginary staggered potentials. We present analytical solutions for the eigenspectrum of this model with both open and closed boundary conditions as well as for the singular-value spectrum. We explicitly demonstrate the proper bulk-boundary correspondence between topological winding numbers in the periodic case and singular values in the open case. We also show that a non-trivial topology leads to protected eigenvalues in the entanglement spectrum. In the $\mathcal{PT}$-symmetric case, we find that the model has a phase where exceptional points become dense in the thermodynamic limit.

## I. INTRODUCTION

It is well known that fundamental principles in quantum mechanics are formulated using Hermitian operators, which assure for example unitary time evolution of closed systems. However, practically, no quantum system is perfectly isolated. The way a system interacts with its environment is often represented through a Master equation like the Lindblad equation [1]. If quantum jumps can be neglected, then the time evolution described by a Lindblad equation can be replaced by an effective non-Hermitian (NH) Hamiltonian [2]. The validity of this effective approach has been verified in different experimental arrangements [3–5]. Strictly speaking, it requires a post-selection of those time evolutions which stay in the considered manifold [6].

Non-Hermitian systems exhibit distinct properties associated with their complex spectrum and the difference between left and right eigenvectors [7]. In this context, as with Hermitian systems, Gaussian non-Hermitian models can offer valuable insights into possible phases and phenomena. In particular, the topological properties of Gaussian non-Hermitian models have been studied and new phenomena have been revealed [8–16]. For example, non-hermiticity leads to a proliferation of possible symmetry classes: the famous Altland-Zirnbauer tenfold classification [17] for Hermitian Gaussian models is replaced by a 38-fold classification [8, 14] in the non-Hermitian domain. The richer classification scheme is a direct consequence of the distinction between transposition and conjugate transposition. Other distinctive properties include the extreme sensitivity of the eigenspectrum to the boundary conditions [15] leading to the skin effect and the breakdown of the conventional bulk-boundary correspondence [10, 12, 18, 19]. Another re-

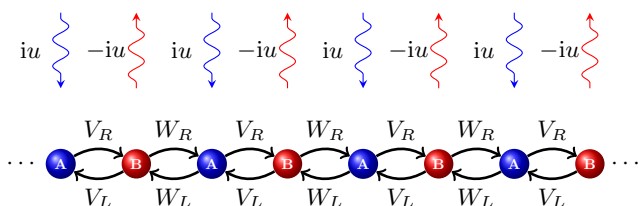

FIG. 1. The considered minimal model has a two-site unit cell, asymmetric intra- ($V_L$, $V_R$) and inter-cell ($W_L$, $W_R$) hopping as well as complex staggered potentials $\pm iu$.

markable aspect of non-Hermitian Hamiltonians is the existence of so-called exceptional points [20] where both eigenvalues and eigenvectors coalesce. Such exceptional points can be detected in experiments [6].

In order to highlight the different phenomena possible in Gaussian non-Hermitian models, we will study a minimal model which allows to turn various symmetries on and off. In the Hermitian case, one of the best-known one-dimensional systems with symmetry-protected topological order is the Su-Schrieffer-Heeger (SSH) model [21] which has chiral symmetry and belongs to the BDI class. This model has alternating strong and weak hoppings and topological edge modes are present if the two sites at the end of an open chain are weakly coupled to their respective neighbors. Here we make the model non-Hermitian by allowing for parity breaking hoppings, i.e., different hopping amplitudes to the left and to the right. In addition, we also introduce staggered imaginary onsite potentials to allow for a non-Hermitian, $\mathcal{PT}$-symmetric limit. Physically, the parity breaking can arise in an effective description of an SSH model with gain and loss. This phenomenon has been investigated in plasmonic chains [22] and photonic lattices [23]. For equal hoppings to the left and to the right, the model is $\mathcal{PT}$ symmetric. The model is depicted in Fig. 1 and can be viewed either as a dimerized Hatano-Nelson model [24] or an SSH chain with asymmetric hopping.

Our paper is organized as follows: In Sec. II, we in-

* mardaniy@myumanitoba.ca
† rodrigo.alves.pimenta@gmail.com
‡ sirker@physics.umanitoba.ca

troduce the model and the difference equations the right and left eigenvectors have to fulfill. In Sec. III, we diagonalize the model for periodic and aperiodic boundary conditions. Furthermore, we determine the parameters for which the excitation gap closes and for which exceptional points do occur. We also introduce and determine the two winding numbers for this system. In Sec. IV, we then analytically study the open boundary case. As expected, the spectrum and the eigenvectors change completely as compared to the closed boundary case. This is also true for the conditions for the closing of the excitation gap and for the location of exceptional points. In Sec. V, we present some of the central results of our paper. We analytically calculate the singular value spectrum of a finite open chain as well as the zero-energy eigenstates of a semi-infinite chain. Based on these results we explicitly establish a bulk-boundary correspondence for this specific model following results recently proven in general in Ref. [25]. In Sec. VI, we then discuss the entanglement entropy when cutting the system in half. The studied model is Gaussian and therefore its correlation matrix can be obtained [26–30]. We find that in topological phases there are topologically protected eigenvalues in the entanglement spectrum. The last section is devoted to a short summary and our conclusions.

## II. THE DIMERIZED HATANO-NELSON MODEL

We consider a model with a two-site unit cell with asymmetric hopping in both intra $(V_L, V_R)$ and inter $(W_L, W_R)$ cells, see Fig. 1. Complex staggered fields $\pm \mathrm{i} u$ act on the sites of the chain. This model can be viewed either as a dimerized Hatano-Nelson model or as a Su-Schrieffer-Heeger (SSH) model with asymmetric hopping. It is described by the Hamiltonian

$$
\begin{aligned}
\mathcal{H} &= \sum_{j=1}^{\lfloor \frac{N-1}{2} \rfloor} W_L c_{2j}^\dagger c_{2j+1} + W_R c_{2j+1}^\dagger c_{2j} \\
&+ \gamma (W_R c_1^\dagger c_L + W_L c_L^\dagger c_1) \\
&+ \sum_{j=1}^{\lfloor \frac{N}{2} \rfloor} V_L c_{2j-1}^\dagger c_{2j} + V_R c_{2j}^\dagger c_{2j-1} \\
&- \mathrm{i} u \sum_{j=1}^{\lfloor \frac{N}{2} \rfloor} c_{2j}^\dagger c_{2j} + \mathrm{i} u \sum_{j=1}^{\lfloor \frac{N+1}{2} \rfloor} c_{2j-1}^\dagger c_{2j-1}, \quad \text{(II.1)}
\end{aligned}
$$

where $c_j$ and $c_j^\dagger$ are fermionic operators satisfying the algebra

$$
\{c_j, c_k\} = \{c_j^\dagger, c_k^\dagger\} = 0, \quad \{c_j, c_k^\dagger\} = \delta_{j,k} \quad \text{(II.2)}
$$

and $\lfloor x \rfloor$ denotes the floor function. By writing the Hamiltonian in the form (II.1), we can consider different boundary conditions (controlled by the parameter $\gamma$) in a uniform fashion. In most parts of the paper, we assume that

the parameters $W_{L,R}$, $V_{L,R}$ and $u$ are real. However, when discussing the exceptional points of this model it makes sense to consider complex parameters as well.

The Hamiltonian (II.1) can be written in the compact form

$$
\mathcal{H} = \boldsymbol{c}^\dagger \mathcal{T}_N \boldsymbol{c} \quad \text{(II.3)}
$$

where $\mathcal{T}_N$ is an asymmetric (almost) tridiagonal matrix with dimension $N \times N$ with matrix elements

$$
\begin{aligned}
t_{2j-1,2j} &= V_L, \quad t_{2j,2j-1} = V_R, \quad t_{2j,2j} = -\mathrm{i} u, \\
t_{2j,2j+1} &= W_L, \quad t_{2j+1,2j} = W_R, \quad t_{2j-1,2j-1} = \mathrm{i} u, \\
t_{1,N} &= \gamma W_R, \quad t_{N,1} = \gamma W_L, \quad \text{(II.4)}
\end{aligned}
$$

and $\boldsymbol{c}^\dagger = (c_1^\dagger, \ldots, c_N^\dagger)$, $\boldsymbol{c} = (c_1, \ldots, c_N)^T$ with $T$ denoting transposition.

To make this paper self-contained, we will recall the diagonalization of the asymmetric SSH chain for both closed $(\gamma = \pm 1)$ and open $(\gamma = 0)$ boundary conditions, referred to respectively as CBC or OBC. In both cases, the diagonalization of $\mathcal{T}_N$ with entries (II.4) follows from difference equations of the spectral problem. Here the lack of hermiticity of the single body Hamiltonian implies that left and right eigenvectors have to be considered. The spectral problems read

$$
\mathcal{T}_N \vec{r} = \epsilon \, \vec{r}, \quad (\mathcal{T}_N)^T \vec{\ell} = \epsilon \, \vec{\ell}, \quad \text{(II.5)}
$$

where $\vec{x} = (x_1, \ldots, x_N)^T$ are right $(x = r)$ and left $(x = \ell)$ eigenvectors of $\mathcal{T}_N$, while $\epsilon$ is an eigenvalue. The components of the right eigenvector satisfy the coupled difference equations

$$
V_L r_{2j} + W_R r_{2j-2} = (\epsilon - \mathrm{i} u) r_{2j-1}, \quad \text{(II.6)}
$$
$$
W_L r_{2j+1} + V_R r_{2j-1} = (\epsilon + \mathrm{i} u) r_{2j}, \quad \text{(II.7)}
$$

with the boundary conditions

$$
r_0 = \gamma r_N, \quad r_{N+1} = \gamma r_1. \quad \text{(II.8)}
$$

Exchanging $R \leftrightarrow L$ one obtains the analogous equations for $\ell_j$, namely

$$
V_R \ell_{2j} + W_L \ell_{2j-2} = (\epsilon - \mathrm{i} u) \ell_{2j-1}, \quad \text{(II.9)}
$$
$$
W_R \ell_{2j+1} + V_L \ell_{2j-1} = (\epsilon + \mathrm{i} u) \ell_{2j}, \quad \text{(II.10)}
$$

with the boundary conditions,

$$
\ell_0 = \gamma \ell_N, \quad \ell_{N+1} = \gamma \ell_1. \quad \text{(II.11)}
$$

For $\gamma = 0$, we consider (II.8, II.11) for both parities of $N$, while for $\gamma \neq 0$ we only consider the case where $N$ is even.

In the following sections, each boundary condition is considered separately.

## III. CLOSED BOUNDARY CONDITIONS

### A. Diagonalization

Let us briefly recapitulate the diagonalization of $\mathcal{T}_N$ for closed boundary conditions (CBC), that is, $\gamma = \pm 1$. In this case, $\mathcal{T}_N$ is a block circulant matrix and can be easily diagonalized. We only consider $N$ even for CBC. The difference equations (II.6,II.7,II.8) and (II.9,II.10,II.11) are easily solved by the ansatz

$$r_{2j-1} = a_R(k)e^{-ikj}, \quad r_{2j} = b_R(k)e^{-ikj}, \quad \text{(III.1)}$$
$$\ell_{2j} = a_L(k)e^{ikj}, \quad \ell_{2j-1} = b_L(k)e^{ikj} \quad \text{(III.2)}$$

where $k$ is a free parameter. The parameters $a_{R,L}(k)$ and $b_{R,L}(k)$ are fixed by the spectral problem of the two band Bloch Hamiltonian

$$H(k) = \begin{pmatrix} iu & H_1(k) \\ H_2(k) & -iu \end{pmatrix}, \quad \text{(III.3)}$$

where

$$H_1(k) = V_L + W_R e^{ik}, \quad \text{(III.4)}$$
$$H_2(k) = V_R + W_L e^{-ik}. \quad \text{(III.5)}$$

That is

$$H(k)|r^\pm(k)\rangle = \epsilon^\pm(k)|r^\pm(k)\rangle,$$
$$\langle \ell^\pm(k)|H(k) = \epsilon^\pm(k)\langle \ell^\pm(k)|, \quad \text{(III.6)}$$

where the quasienergies are given by

$$\epsilon^\pm(k) = \pm\sqrt{H_1(k)H_2(k) - u^2}, \quad \text{(III.7)}$$

and the left and right eigenvectors by

$$|r^\pm(k)\rangle = \frac{1}{\sqrt{N^\pm}}\begin{pmatrix} a_R^\pm(k) \\ b_R^\pm(k) \end{pmatrix}$$
$$= \frac{1}{\sqrt{N^\pm}}\begin{pmatrix} iu \pm \sqrt{H_1(k)H_2(k) - u^2} \\ H_2(k) \end{pmatrix}, \quad \text{(III.8)}$$

$$\langle \ell^\pm(k)| = \frac{1}{\sqrt{N^\pm}}\left( a_L^\pm(k), \; b_L^\pm(k) \right),$$
$$= \frac{1}{\sqrt{N^\pm}}\left( iu \pm \sqrt{H_1(k)H_2(k) - u^2}, \; H_1(k). \right)$$
$$\text{(III.9)}$$

Normalizing left and right eigenvectors according to the biorthogonality condition

$$\langle \ell^\epsilon(k)|r^{\epsilon'}(k)\rangle = \delta_{\epsilon,\epsilon'}, \quad \text{(III.10)}$$

fixes the normalization constant

$$N^\pm = 2\left( H_1(k)H_2(k) - u^2 \pm iu\sqrt{H_1(k)H_2(k) - u^2} \right). \quad \text{(III.11)}$$

To conclude the solution, the parameter $k$ is quantized by (II.8) and given by

$$k_m^{(\text{PBC})} = \frac{2\pi(2m)}{N}, \quad m = 1, \ldots, \frac{N}{2}, \quad \text{(III.12)}$$

for periodic boundaries and

$$k_m^{(\text{APBC})} = \frac{2\pi(2m+1)}{N}, \quad m = 1, \ldots, \frac{N}{2}, \quad \text{(III.13)}$$

for antiperiodic boundaries.

### B. Gap closing and exceptional points

The bands (III.7) for the asymmetric SSH chain are in general complex, and a band touching requires the vanishing of both the real and the imaginary parts of the complex gap

$$\Delta_c = (\epsilon^+(k) - \epsilon^-(k))/2 = \sqrt{H_1(k)H_2(k) - u^2}. \quad \text{(III.14)}$$

The closing of both the real and the imaginary parts is well captured by the non-Hermitian gap [31]

$$\Delta = \min_k |\Delta_c(k)|. \quad \text{(III.15)}$$

The complex gap (III.14) closes when,

$$H_1(k)H_2(k) - u^2 = 0. \quad \text{(III.16)}$$

For each value of $k$ quantized by (III.12,III.13), the condition (III.16) leads to constraints in the model parameters and can determine a rich variety of phase boundaries. Furthermore, it follows immediately from Eqs. (III.8,III.9) that at points where the complex gap (III.16) vanishes the eigenvectors $\{|r^+(k)\rangle, |r^-(k)\rangle\}$ and also $\{\langle \ell^+(k)|, \langle \ell^-(k)|\}$ coalesce. Gap closing points are therefore also always exceptional points. Exceptional points and their relation with the discriminant of the characteristic polynomial of the hopping matrix $\mathcal{T}_N$ are discussed further in App. B.

To analyze the manifold of exceptional points given by Eq. (III.16) it is convenient to first set $u = 0$. In this case, the system has sublattice symmetry $\sigma^z H(k)\sigma^z = -H(k)$.[1] Then, for every allowed quantized $k_m$ value and relaxing the requirement of real couplings, the solutions of $H_1 = 0$ or $H_2 = 0$ are

$$\left(\frac{V_L}{W_R}\right)_{\text{EP}} = -e^{ik_m} \quad \text{or} \quad \left(\frac{W_L}{V_R}\right)_{\text{EP}} = -e^{ik_m}. \quad \text{(III.17)}$$

where $k_m$ is given by (III.12,III.13). It follows that for most values of $k_m$ the coupling parameters which lead to a gap closing/exceptional point are complex. For finite $N$ and periodic boundaries, one can see that $m = N/2$ ($k = 2\pi$) and $m = N/4$ ($k = \pi$, if $N/2$ is even) are the only possible values of $k$ that leads to real ratios,

$$\left(\frac{V_L}{W_R}\right)_{\text{EP}} = \pm 1 \quad \text{or} \quad \left(\frac{W_L}{V_R}\right)_{\text{EP}} = \pm 1. \quad \text{(III.18)}$$

_______

[1] In non-Hermitian systems, chiral and sublattice symmetries are not equivalent [14]. Indeed for $u = 0$ the chiral symmetry $\sigma^z H(k)^\dagger \sigma^z = -H(k)$ is not satisfied since $\sigma^z H(k)^\dagger \sigma^z = -H(k)^\dagger$.

Fixing $V_R = W_R = 1$ (no loss in generality if $u = 0$), these exceptional lines can be visualized as a function of $V_L$ and $W_L$, see the left panel in Fig. 2. For finite $N$ and antiperiodic boundaries, the ratios (III.17) are real for $m = 1/2(N/2-1)$ ($k = \pi$, if $N/2-1$ is even). In the thermodynamic limit, the unit circles are filled and real ratios occur at $k = \pi$ or $k = 2\pi$ for both periodic and antiperiodic boundary conditions. We remark that the sublattice-symmetric model has been widely considered in the literature, see *e.g.* Refs. [11, 13, 32–35]. Here we emphasize that the gap closing points and exceptional points coincide and that they can also occur in the complex parameter plane for every allowed, quantized value $k_m$.

Now we consider a nonzero real field $u$. The field couples $H_1(k)$ and $H_2(k)$ and Eq. (III.16) no longer factorizes, giving a constraint for all model parameters for every possible value of $k$. Eq. (III.16) can be solved for any of the model parameters, leading in general to a complex number. As an example, in the central panel of Fig. 2, the non-Hermitian gap (III.15) is shown for fixed parameters $V_R = W_R = 1$ and $u = 0.5$ as a function of $V_L$ and $W_L$. In this case, we can observe the merging of the zero field phases $(\nu_1, \nu_2) = (0,0)$ and $(1,-1)$ (we discuss the meaning of these numbers below) into a single phase as well as the bending of the phases boundaries.

Another interesting case to consider is the $\mathcal{PT}$ symmetric case with $V_L = V_R = V$ and $W_L = W_R = W$ but the real field $u$ is arbitrary, such that $\sigma^x H(k) \sigma^x = H(k)^*$. In this case, the constraint (III.16) reduces to,

$$V^2 + W^2 + 2VW \cos(k_m) - u^2 = 0. \qquad \text{(III.19)}$$

To visualize the gap closing, we can set $u = 1$ (no loss of generality) and plot the gap (III.15) as a function of $V$ and $W$, see the right panel of Fig. 2. The dark blue lines mark the the gap closing/ exceptional points, which only occur within the regions $|V - W| < 1 < |V + W|$ or $|V + W| < 1 < |V - W|$. In the thermodynamic limit, this region is gapless and filled with exceptional points. It thus constitutes an entire exceptional phase.

We end this subsection remarking that in addition to the collision of quasienergies $\Delta_c = 0$, another type of degeneracy in the spectrum can also be found. This degeneracy occurs within each band $\epsilon^\pm$ for different quasimomentum $k$. Namely, one can impose

$$H_1(k)H_2(k) = H_1(k')H_2(k') \qquad \text{(III.20)}$$

which gives another constraint on the model parameters. However, these degeneracies do not lead to exceptional points, as the eigenvectors do not coalesce, recall (III.8,III.9). This type of degeneracy, however, does appear as a factor in the discriminant of the characteristic polynomial of $\mathcal{T}_N$, see further remarks on this point in App. B, in particular Eq. (B.3).

## C. Winding numbers

The eigenvalues of a non-Hermitian Hamiltonian are in general complex, and can trace a closed path around an arbitrary reference complex energy $E_B$, in which case a winding number can be assigned [18, 32, 36–38],

$$I_1 = \frac{1}{2\pi i} \int_0^{2\pi} dk \, \log \det(H(k) - E_B), \qquad \text{(III.21)}$$

provided that $H(k)$ has a point gap, that is, $\det(H(k) - E_B) \neq 0$. For concreteness we will chose $E_B = 0$. Observe that the definition (III.21) is universal since no symmetry requirements are made. Therefore, one dimensional non-Hermitian systems—in contrast to Hermitian ones—can have non-trivial topology in the absence of any spatial or non-spatial symmetries.

If symmetries are present, then they lead to additional topological invariants [14]. So far, mostly the case of a two-band model with sub-lattice symmetry has been considered [11, 38–42]. The Hamiltonian of such a model takes the form

$$H(k) = \begin{pmatrix} 0 & \hat{H}_1(k) \\ \hat{H}_2(k) & 0 \end{pmatrix}. \qquad \text{(III.22)}$$

Using the polar decomposition $\hat{H}_j(k) = d_j(k)q_j(k)$ with $q_j(k)$ unitary and $d_j(k)$ positive semi-definite, one can define two winding numbers

$$\nu_j = \frac{1}{2\pi i} \int dk \, \text{tr} \left( q_j^\dagger \partial_k q_j \right) = \frac{1}{2\pi i} \int dk \, \partial_k \log \det q_j. \qquad \text{(III.23)}$$

Instead of the winding numbers $\nu_{1,2}$, one can equivalently consider [13, 43]

$$I_1 = \nu_1 + \nu_2, \qquad \text{(III.24)}$$

which is consistent with the definition of $I_1$ in Eq. (III.21), and

$$I_2 = \nu_2 - \nu_1 \qquad \text{(III.25)}$$

as independent winding numbers. For our model with $u = 0$, we have $H_j(k) = |H_j(k)|e^{i\phi_j}$ and evaluating the integrals (III.24,III.25) leads to

$$I_1(u = 0) = \begin{cases} -1, & \text{if} \quad \left|\frac{W_L}{V_R}\right| > 1 \wedge \left|\frac{V_L}{W_R}\right| > 1 \\ 1, & \text{if} \quad \left|\frac{W_L}{V_R}\right| < 1 \wedge \left|\frac{V_L}{W_R}\right| < 1 \\ 0, & \text{otherwise} \end{cases} \qquad \text{(III.26)}$$

and

$$I_2 = \begin{cases} -2 & \text{if} \quad \left|\frac{V_L}{W_R}\right| < 1 \wedge \left|\frac{W_L}{V_R}\right| > 1 \\ 0 & \text{if} \quad \left|\frac{V_L}{W_R}\right| > 1 \wedge \left|\frac{W_L}{V_R}\right| < 1 \\ -1 & \text{otherwise.} \end{cases} \qquad \text{(III.27)}$$

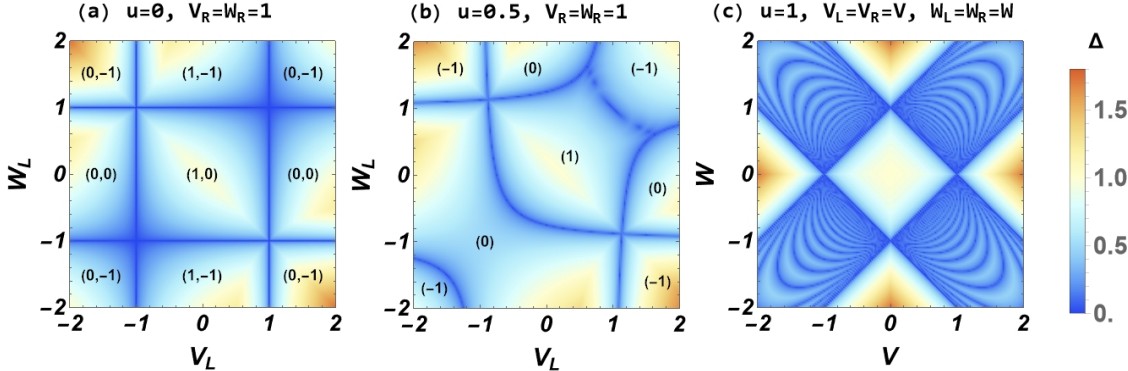

FIG. 2. The non-Hermitian gap $\Delta$ as a function of the model parameters for $N = 100$ and periodic boundary conditions. In panels (a) and (b) we set $V_R = W_R = 1$ and $u = 0$ and $u = 0.5$ respectively. In panel (c) we consider $V_L = V_R = V$, $W_L = W_R = W$ and the field $u = 1$. The numbers in panel (a) indicate the two winding numbers $(\nu_1, \nu_2)$, the numbers in panel (b) the winding number $I_1$.

Clearly, the winding numbers are not defined when $\left|\frac{V_L}{W_R}\right| = \left|\frac{W_L}{V_R}\right| = 1$ which are precisely the gapless/exceptional lines discussed in the previous subsection. The two winding numbers $\nu_{1,2}$ are shown in the left panel of Fig. 2 for parameters $V_R = W_R = 1$ (no loss of generality) in each phase as a function of $V_L$ and $W_L$.

Next, we briefly discuss the $\mathcal{PT}$-symmetric case. The Hamiltonian (III.3) for $V_L = V_R = V$ and $W_L = W_R = W$ satisfies the $\mathcal{PT}$ symmetry $\sigma^x H(k)\sigma^x = H(k)^*$ as well as chiral symmetry $\sigma^z H(k)^\dagger \sigma^z = -H(k)$, and this case is part of the AZ symmetry class AIII [14]. The phase diagram for $u = 1$ is shown in Fig. 2(c) and consists of gapped phases and phases where exceptional points become dense in the thermodynamic limit. These phases will be discussed further in Sec. VI when we analyze the entanglement spectrum.

## IV. OPEN BOUNDARY CONDITIONS

### A. Diagonalization

Let us also recall the diagonalization of $\mathcal{T}_N$ (II.4) for open boundary conditions (OBC), that is, $\gamma = 0$. In this case, $\mathcal{T}_N$ is an asymmetric tridiagonal 2-Toeplitz matrix whose eigenvalue problem was solved from different perspectives in Refs. [44, 45]. In App. A, we provide some details on both approaches. The connection between these two approaches was pointed out in Ref. [46] and the method [45] was previously applied to the Hermitian SSH chain in Refs. [47, 48].

The solution of the open boundary case can also be obtained by a simple transformation to the standard (symmetric) SSH chain [12], namely,

$$S\mathcal{T}_N S^{-1} = \mathcal{T}_N^{SSH} \qquad (IV.1)$$

where $S$ is a diagonal matrix with entries

$$s_{2j-1} = \frac{V_R^{\frac{j-1}{2}} W_R^{\frac{j-1}{2}}}{V_L^{\frac{j-1}{2}} W_L^{\frac{j-1}{2}}}, \qquad s_{2j} = \frac{V_R^{\frac{j}{2}} W_R^{\frac{j}{2}}}{V_L^{\frac{j}{2}} W_L^{\frac{j}{2}}} \frac{W_L}{W_R}, \qquad (IV.2)$$

and $\mathcal{T}_N^{SSH}$ is a symmetric hopping matrix of the standard SSH chain with effective couplings,

$$V_{\text{eff}} = \sqrt{V_L V_R}, \qquad W_{\text{eff}} = \sqrt{W_L W_R}. \qquad (IV.3)$$

The quasienergies are then given by

$$\epsilon^\pm = \pm\sqrt{\tilde{H}_1(\theta)\tilde{H}_2(\theta) - u^2}, \qquad (IV.4)$$

where

$$\tilde{H}_1(\theta) = V_L(e^{i\theta/2} + e^{-i\theta/2}\delta^{-1/2}), \qquad (IV.5)$$

$$\tilde{H}_2(\theta) = V_R(e^{-i\theta/2} + e^{i\theta/2}\delta^{-1/2}), \qquad (IV.6)$$

with

$$\sqrt{\delta} = \frac{\sqrt{V_L}\sqrt{V_R}}{\sqrt{W_L}\sqrt{W_R}}, \qquad (IV.7)$$

and $\theta$ is a parameter that is given by different expressions depending on the parity of $N$. We remark that, for odd $N$, there is an isolated root $\epsilon = iu$ which is not contained in (IV.4).

For odd $N$, the parameter $\theta$ is simply

$$\theta_m^{(OBC)} = \frac{2\pi m}{N + 1}, \qquad m = 1, \ldots, \frac{N-1}{2}, \qquad (IV.8)$$

which gives $N - 1$ eigenvalues from (IV.4) and (IV.8). Together with the isolated root $\epsilon = iu$, one has all the $N$ eigenvalues of the tridiagonal matrix when $\gamma = 0$.

For even $N$, the parameter $\theta$ is determined via the transcendental equation

$$\sqrt{\delta}\sin\left(\left(\frac{N}{2} + 1\right)\theta\right) + \sin\left(\frac{N}{2}\theta\right) = 0. \qquad (IV.9)$$

We remark that the transcendental equation (IV.9) has the same form as the transcendental equation of the quantum Ising chain with transverse field $\lambda = \sqrt{\delta}$ and $L/2$ sites [49]. Meanwhile, the eigenvector components can be written as

$$r_{2n} = \frac{V_R^{\frac{n-1}{2}} W_R^{\frac{n-1}{2}}}{V_L^{\frac{n-1}{2}} W_L^{\frac{n-1}{2}}} (\epsilon - iu) \frac{\sin(n\theta)}{\sin(\theta)}, \qquad (IV.10)$$

$$r_{2n-1} = \frac{V_R^{\frac{n-1}{2}} W_R^{\frac{n-1}{2}}}{V_L^{\frac{n-1}{2}} W_L^{\frac{n-1}{2}}} \frac{\left(\sin(n\theta) + \delta^{-\frac{1}{2}} \sin((n-1)\theta)\right)}{V_L^{-1} \sin(\theta)}, \qquad (IV.11)$$

and

$$\ell_{2n} = \frac{V_L^{\frac{n-1}{2}} W_L^{\frac{n-1}{2}}}{V_R^{\frac{n-1}{2}} W_R^{\frac{n-1}{2}}} (\epsilon - iu) \frac{\sin(n\theta)}{\sin(\theta)}, \qquad (IV.12)$$

$$\ell_{2n-1} = \frac{V_L^{\frac{n-1}{2}} W_L^{\frac{n-1}{2}}}{V_R^{\frac{n-1}{2}} W_R^{\frac{n-1}{2}}} \frac{\left(\sin(n\theta) + \delta^{-\frac{1}{2}} \sin((n-1)\theta))\right)}{V_R^{-1} \sin(\theta)}. \qquad (IV.13)$$

Note that all the eigenstates are, in general, localized at the boundaries for the open case in contrast to the closed case where they are extended Bloch waves. More specifically, the right eigenstates are localized at the left (right) boundary for $\Gamma < 1$ ($\Gamma > 1$) with $\Gamma = V_R W_R / V_L W_L$. Conversely, the left eigenstates are localized at the right (left) boundary for $\Gamma < 1$ ($\Gamma > 1$). There is no localization if $\Gamma = 1$. The localization of all the eigenstates when switching from closed to open boundary conditions is called the non-Hermitian skin effect.

### B. Gap closing and exceptional points

Similar to the case with closed boundary conditions, one can define a complex gap

$$\Delta_c^{(OBC)} = (\epsilon^+(\theta) - \epsilon^-(\theta))/2 = \sqrt{\tilde{H}_1(\theta)\tilde{H}_2(\theta) - u^2}$$
$$= \sqrt{V_L V_R + W_L W_R + 2\sqrt{V_L}\sqrt{V_R}\sqrt{W_L}\sqrt{W_R}\cos\theta - u^2} \qquad (IV.14)$$

as well as a minimal gap

$$\Delta^{(OBC)} = \min_\theta |\Delta_c^{(OBC)}(\theta)|, \qquad (IV.15)$$

where $\theta$ is given by (IV.8) or by a solution of (IV.9) depending on the parity of $N$. A different analysis of the complex gap is required for each parity of $N$.

#### 1. Odd N

Let us start with odd $N$ where the parameter $\theta \to \theta_m$ is explicitly given by (IV.8). Also, for clarity, let us consider

first $u = 0$. In this case, the complex gap (IV.14) will vanish if $\tilde{H}_1(\theta_m)\tilde{H}_2(\theta_m) = 0$, which happens if any of the hopping parameters vanishes or if

$$\left(\sqrt{\delta}\right)_{\text{EP}} = -e^{-i\theta_m} \quad \text{or} \quad \left(\sqrt{\delta}\right)_{\text{EP}} = -e^{i\theta_m}. \qquad (IV.16)$$

Therefore, for non-vanishing hopping parameters, there are $N - 1$ exceptional points. For finite $N$, these points always have an imaginary part. In the thermodynamic limit, however, the exceptional points will converge to the real axis

$$\lim_{N\to\infty} \left(\sqrt{\delta}\right)_{\text{EP}} = \pm 1 \qquad (IV.17)$$

in the cases when $\theta_m \to 0$ or $\theta_m \to \pi$.

Similarly to the closed boundary case, turning on the field $u$ couples $\tilde{H}_1(\theta_m)$ and $\tilde{H}_2(\theta_m)$. We observe that introducing the rescaled field,

$$\bar{u} = u/(\sqrt{W_L}\sqrt{W_R}) \qquad (IV.18)$$

one can write

$$\tilde{H}_1(\theta_m)\tilde{H}_2(\theta_m) - u^2 = \sqrt{V_L}\sqrt{V_R}\sqrt{W_L}\sqrt{W_R}$$
$$\times \left(\sqrt{\delta} + \frac{1}{\sqrt{\delta}}(1 - \bar{u}^2) + 2\cos(\theta_m)\right). \qquad (IV.19)$$

The minimal gap (IV.15) as a function of $\bar{u}$ and $\delta$ is shown in Fig. 3. The exceptional points are therefore now given

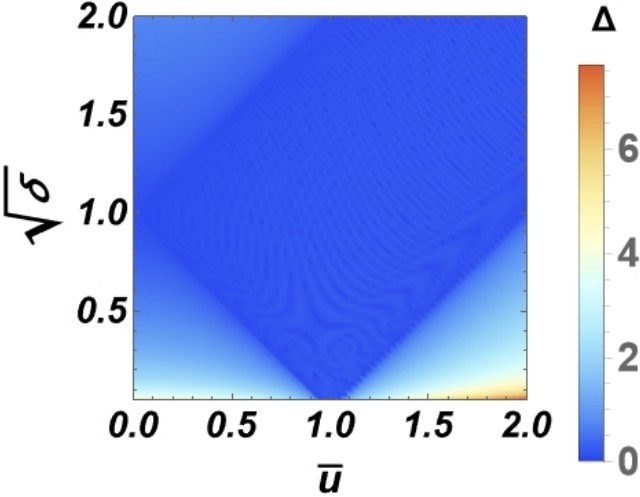

FIG. 3. The rescaled minimal gap $\Delta$ as a function of $\sqrt{\delta}$ and $\bar{u}$ for $N = 101$ and open boundary conditions.

by

$$\left(\sqrt{\delta}\right)_{\text{EP}} = -\cos(\theta_m) \pm \sqrt{\bar{u}^2 - \sin^2(\theta_m)}. \qquad (IV.20)$$

Interestingly, the expression (IV.20) implies that real exceptional points do exist for finite $N$ provided that $\bar{u} \geq |\sin(\theta_m)|$ or $\bar{u} \leq -|\sin(\theta_m)|$. In the thermodynamic

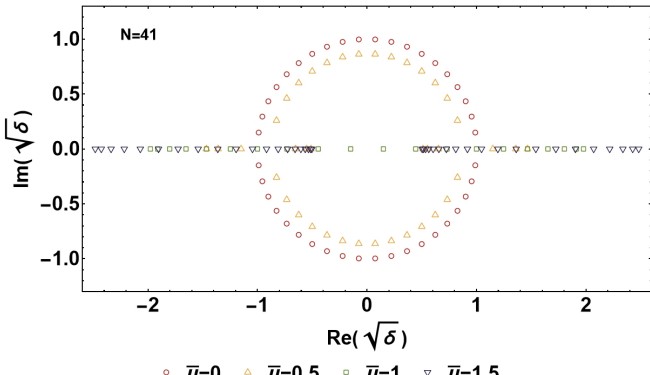

FIG. 4. Loci of exceptional points $\left(\sqrt{\delta}\right)_{\text{EP}}$ for $N = 41$ and different rescaled fields $\bar{u}$.

limit, critical fields $\bar{u} \to \pm 1$ emerge when $\theta_m$ approaches $0$ or $\pi$. In Fig. 4 some examples of the loci of exceptional points for different values of the rescaled field $\bar{u}$ are shown.

Recall that when $N$ is odd, there is, in addition, an isolated root $\epsilon = iu$. We can impose that this root coincides with the quasienergies (IV.4), that is, $iu = \epsilon^{\pm}$. For non-zero hopping parameters, this implies again the condition (IV.16), but now associated with the degenerate quasienergy $iu$.

### 2. Even N

Similar to the previously discussed case of odd $N$, the complex gap (IV.14) will vanish if

$$\sqrt{\delta} + \frac{1}{\sqrt{\delta}}(1 - \bar{u}^2) + 2\cos(\theta) = 0 \qquad \text{(IV.21)}$$

but with $\theta$ now given by by (IV.9). Since Eq. (IV.9) also involves $\sqrt{\delta}$, one has to consider both equations simultaneously. Note that both conditions are satisfied if any of the hopping parameters vanishes similar to the case of $N$ odd. Otherwise, after some manipulation, the intersection of (IV.21) and (IV.9) leads to,

$$\sin(\theta) \pm \bar{u}\sin\left(\left(\frac{N}{2}+1\right)\theta\right) = 0, \qquad \text{(IV.22)}$$

whose solutions $\theta = \theta_{EP}$ determine exceptional values of $\sqrt{\delta}$ from either (IV.21) or (IV.9). That is, choosing (IV.9), the exceptional points are given by,

$$\left(\sqrt{\delta}\right)_{\text{EP}} = -\frac{\sin\left(\frac{N}{2}\theta_{\text{EP}}\right)}{\sin\left(\left(\frac{N}{2}+1\right)\theta_{\text{EP}}\right)}. \qquad \text{(IV.23)}$$

The transcendental equations (IV.22) can be solved numerically for different rescaled fields $\bar{u}$. We observe that each sign in (IV.22) has in general $N/2$ solutions with $\text{Re}(\theta_{\text{EP}}) \geq 0$ which determine $N$ exceptional points

(other solutions with $\text{Re}(\theta_{\text{EP}}) < 0$ do not produce new exceptional points). An exception is the case $\bar{u} = 1$ where the number of solutions of Eq. (IV.22) with a negative sign has one less solution. Nevertheless, all distinct exceptional points are obtained. We show some examples in Fig. 5.

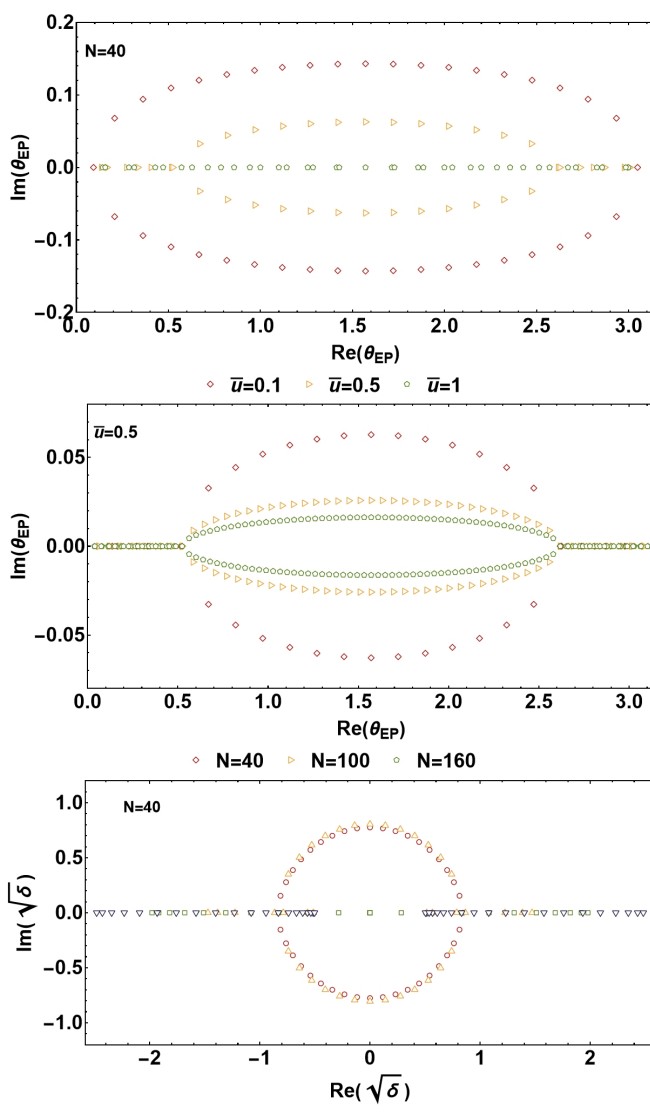

FIG. 5. In the upper panel, solutions of the transcendental equation (IV.22) for different fields $\bar{u}$ and $N = 40$ are shown. In the middle panel, we fix $\bar{u} = 1/2$ and consider different lattice sizes. In the bottom panel, we show the exceptional points associated with (IV.22).

There is an additional possibility to cancel the discriminant, first discovered in the study of exceptional points of the Baxter free parafermionic model [50]. It consists in imposing that the derivative of (IV.9) vanishes,

$$\sqrt{\delta}(N+2)\cos\left(\left(\frac{N}{2}+1\right)\theta\right) + N\cos\left(\frac{N}{2}\theta\right) = 0 \qquad \text{(IV.24)}$$

in addition to (IV.9). The intersection of (IV.24) and (IV.9) then gives [50],

$$\sin((N+1)\theta) - (N+1)\sin(\theta) = 0 \qquad \text{(IV.25)}$$

whose solutions $\theta = \theta'_{\text{EP}}$ give extra exceptional points

$$\left(\sqrt{\delta}\right)_{\text{EP}} = -\frac{\sin\left(\frac{N}{2}\theta'_{\text{EP}}\right)}{\sin\left((\frac{N}{2}+1)\theta'_{\text{EP}}\right)}. \qquad \text{(IV.26)}$$

The transcendental equation (IV.25) is clearly independent of the field $u$. It can be solved numerically; there are $N-2$ solutions [50] with $\text{Re}(\theta'_{\text{EP}}) \geq 0$ which fully describe the field independent part of the discriminant. The numerical solution for different lattice sizes is shown in the upper panel of Fig. 6. Note that in this case the associated repeated quasienergy is not zero. Instead, they trace interesting curves in the complex plane, see the examples shown in the bottom panel of Fig. 6.

## V. BULK-BOUNDARY CORRESPONDENCE

It is well-known that the standard bulk-boundary correspondence for topological Hermitian systems breaks down in the non-Hermitian case. Mathematically, the fundamental issue is that the eigenvalues of a finite non-Hermitian matrix do, in general, not converge to the eigenvalues of the semi-infinite case. This is similar to a discontinuous point. More specifically, in the Hermitian case we can connect the topological invariant calculated for a periodic system with the number of edge states in a finite system. The energy of these edge states converges exponentially to zero with system size. This is what is known as the standard bulk-boundary correspondence.

In the non-Hermitian case, the eigenspectrum cannot be used to construct a bulk boundary correspondence for a finite system. Instead, it is known from Toeplitz theory that the singular value spectrum has to be used. The K-splitting theorem states [51] that a system with winding number $I$ has $K \geq |I|$ singular values which will converge to zero with increasing system size and which belong to states which are exponentially localized at the boundary. In the semi-infinite case these states become exact eigenstates with exactly zero energy but for finite system size they only get mapped exponentially close to zero by the Hamiltonian but are not exact eigenstates. I.e., in a finite non-Hermitian system the zero energy edge modes are 'hidden', in general, and constitute very long-lived states. The proper bulk-boundary correspondence for non-Hermitian systems was recently fully explored in Ref. [25]. We note that this formalism does encompass the Hermitian case because for a Hermitian matrix the singular values are just the absolute values of the eigenvalues. Here we want to demonstrate the bulk-boundary correspondence explicitly for the model under consideration.

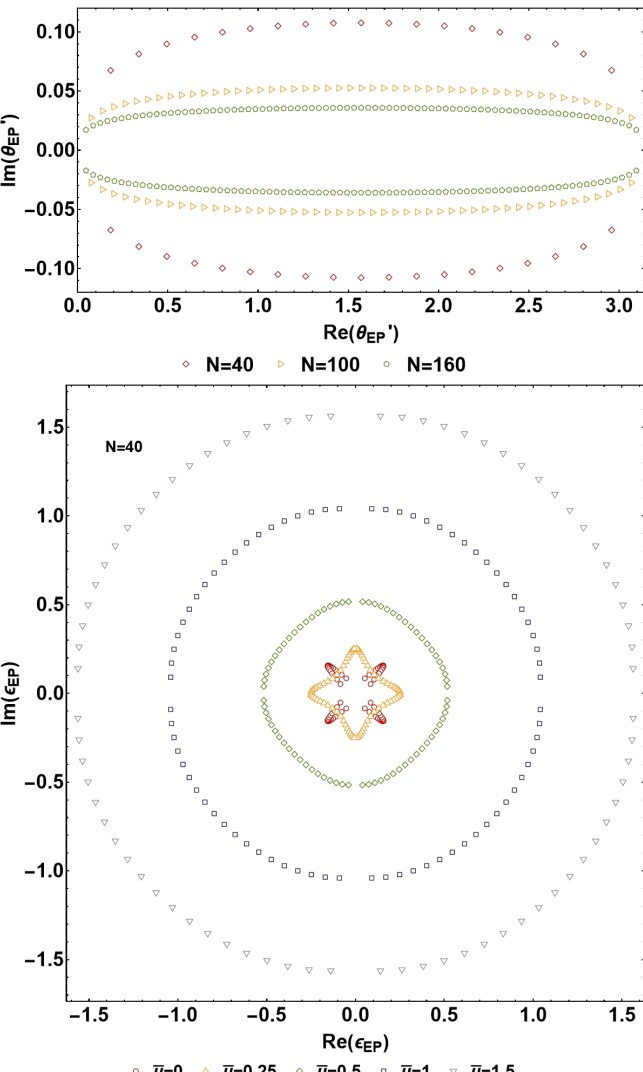

FIG. 6. (a) The solution of Eq. (IV.26) leads to additional, $\bar{u}$ independent exceptional points. (b) The corresponding energies $\varepsilon_{EP}$ lie along non-trivial curves in the complex plane.

### A. Singular values

To do so, we start by computing the eigenvalues of

$$\Sigma_N = \mathcal{T}_N \mathcal{T}_N^\dagger \qquad \text{(V.1)}$$

when $\gamma = u = 0$, that is, the open case with zero field. The singular values $\sigma$ of $\mathcal{T}_N$ are given by the square roots of the eigenvalues of $\Sigma_N$ which is a symmetric band matrix with elements,

$$(\Sigma_N)_{2j-1,2j+1} = (\Sigma_N)_{2j-1,2j-3} = V_L W_R,$$
$$(\Sigma_N)_{2j,2j-2} = (\Sigma_N)_{2j,2j+2} = W_L V_R,$$
$$(\Sigma_N)_{2j-1,2j-1} = V_L^2 + W_R^2, \quad (\Sigma_N)_{2j,2j} = W_L^2 + V_R^2$$
$$(\Sigma_N)_{1,1} = V_L^2, \quad (\Sigma_N)_{L,L} = \begin{cases} V_R^2, & \text{even L} \\ W_R^2, & \text{odd L} \end{cases}. \quad \text{(V.2)}$$

This means that $\Sigma_N$ represents a hopping model with next-nearest neighbour interaction.

The spectral problem $(\Sigma_N)\vec{s} = \sigma^2 \vec{s}$ reads,

$$s_{2j-3} + s_{2j+1} = \frac{\sigma^2 - (V_L^2 + W_R^2)}{V_L W_R} s_{2j-1}$$

$$s_{2j-2} + s_{2j+2} = \frac{\sigma^2 - (W_L^2 + V_R^2)}{W_L V_R} s_{2j} \qquad \text{(V.3)}$$

with boundaries fixed by the last line in (V.2). Using the method reviewed in App. A, we find that there are two solutions to the difference equations (V.3).

In the first (second) solution the even (odd) components of the vector $\vec{s}$ are zero $s_{2j} = 0$ ($s_{2j-1} = 0$) and the left boundary fixes the ratio $s_3/s_1$ ($s_4/s_2$). The first solution has singular values

$$\sigma_\phi^2 = V_L^2 + W_R^2 + 2V_L W_R \cos(\phi) \qquad \text{(V.4)}$$

and vector components,

$$\frac{s_{2j-1}}{s_1} = \frac{W_R}{V_L}\frac{\sin((j-1)\phi)}{\sin(\phi)} + \frac{\sin(j\phi)}{\sin(\phi)}, \quad j \geq 3, \quad \text{(V.5)}$$

$$\frac{s_3}{s_1} = \frac{W_R}{V_L} + 2\cos(\phi), \qquad \text{(V.6)}$$

where $\phi$ satisfies the transcendental equation,

$$\sin\left(\frac{N}{2}\phi\right)\left(\frac{W_R}{V_L} + 2\cos(\phi)\right) - \sin\left(\left(\frac{N}{2} - 1\right)\phi\right) = 0. \qquad \text{(V.7)}$$

The second solution has eigenvalues

$$\sigma_\omega^2 = W_L^2 + V_R^2 + 2W_L V_R \cos(\omega) \qquad \text{(V.8)}$$

and vector components,

$$\frac{s_{2j}}{s_2} = \frac{\sin(j\omega)}{\sin(\omega)}, \quad j \geq 3, \qquad \text{(V.9)}$$

$$\frac{s_4}{s_2} = 2\cos(\omega) \qquad \text{(V.10)}$$

with the parameter $\omega$ satisfying,

$$\sin\left(\left(\frac{N}{2} + 1\right)\omega\right) + \frac{W_L}{V_R}\sin\left(\frac{N}{2}\omega\right) = 0. \qquad \text{(V.11)}$$

The spectrum of (V.1) is given by the union of $\sigma_\phi^2$ and $\sigma_\omega^2$. Remarkably, the pairs of variables $\{V_L, W_R\}$ and $\{V_R, W_L\}$ decouple from each other. This means that the characteristic polynomial associated with (V.1) factorizes into two independent factors depending solely on each of these pairs.

The transcendental equations (V.7) and (V.11) can be solved numerically. We find that (V.7) has $N/2$ real solutions in the interval $0 < \phi < \pi$ when $|V_L/W_R| \geq 1$. When $|V_L/W_R| < 1$ a complex conjugate pair emerges near $\pi$.

This complex root produces an exponentially small singular value $\sigma_\phi$. Similarly, we find that (V.11) has $N/2$ real solutions in the interval $0 < \omega < \pi$ when $|V_R/W_L| \geq 1$ and $N/2 - 1$ real solutions plus a complex conjugate pair solution when $|V_R/W_L| < 1$. In Fig. 7, we plot the smallest two singular values coming from each transcendental equation considering a lattice with $N = 100$ sites.

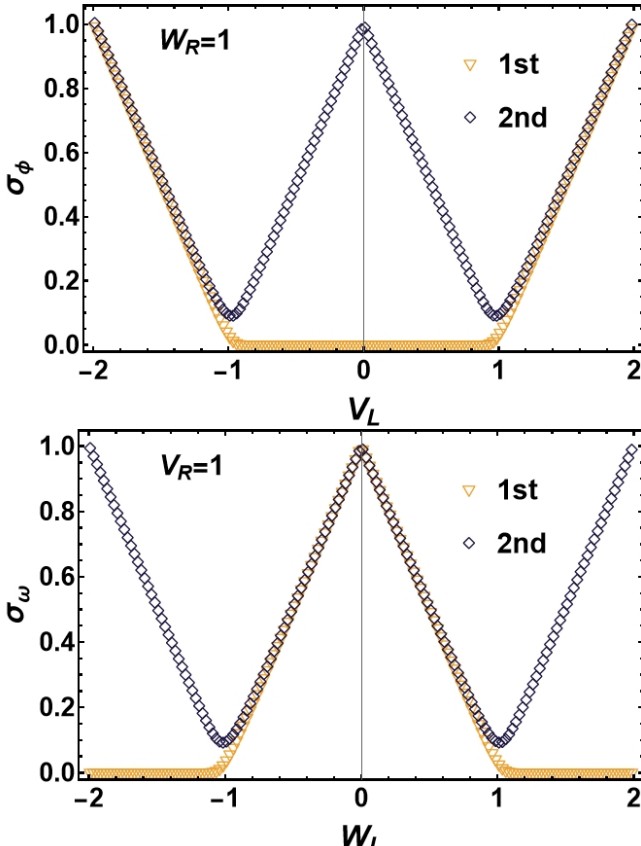

FIG. 7. The two smallest singular values from Eq. (V.4) (upper panel) and from Eq. (V.8) (lower panel). In both cases the corresponding transcendental equation is solved numerically.

The Bloch Hamiltonian for the corresponding periodic problem with $u = 0$ has block structure, see Eq. (III.22). There are therefore two winding numbers $\nu_{1,2}$ defined in Eq. (III.23). The k-splitting theorem then predicts that there are $K \geq |\nu_1| + |\nu_2|$ singular values $\sigma$ with $\lim_{N\to\infty} \sigma = 0$ [25, 51]. This is fully consistent with our results, compare Fig. 7 with the phase diagram in the left panel of Fig. 2.

## B. Topological protection and hidden zero modes

In the limit of a semi-infinite chain, the system will have $K \geq |\nu_1| + |\nu_2|$ topologically protected edge modes with exactly zero energy. However, these states will only be eigenstates in this limit but not for a finite system. For a finite system, the singular values have to be considered

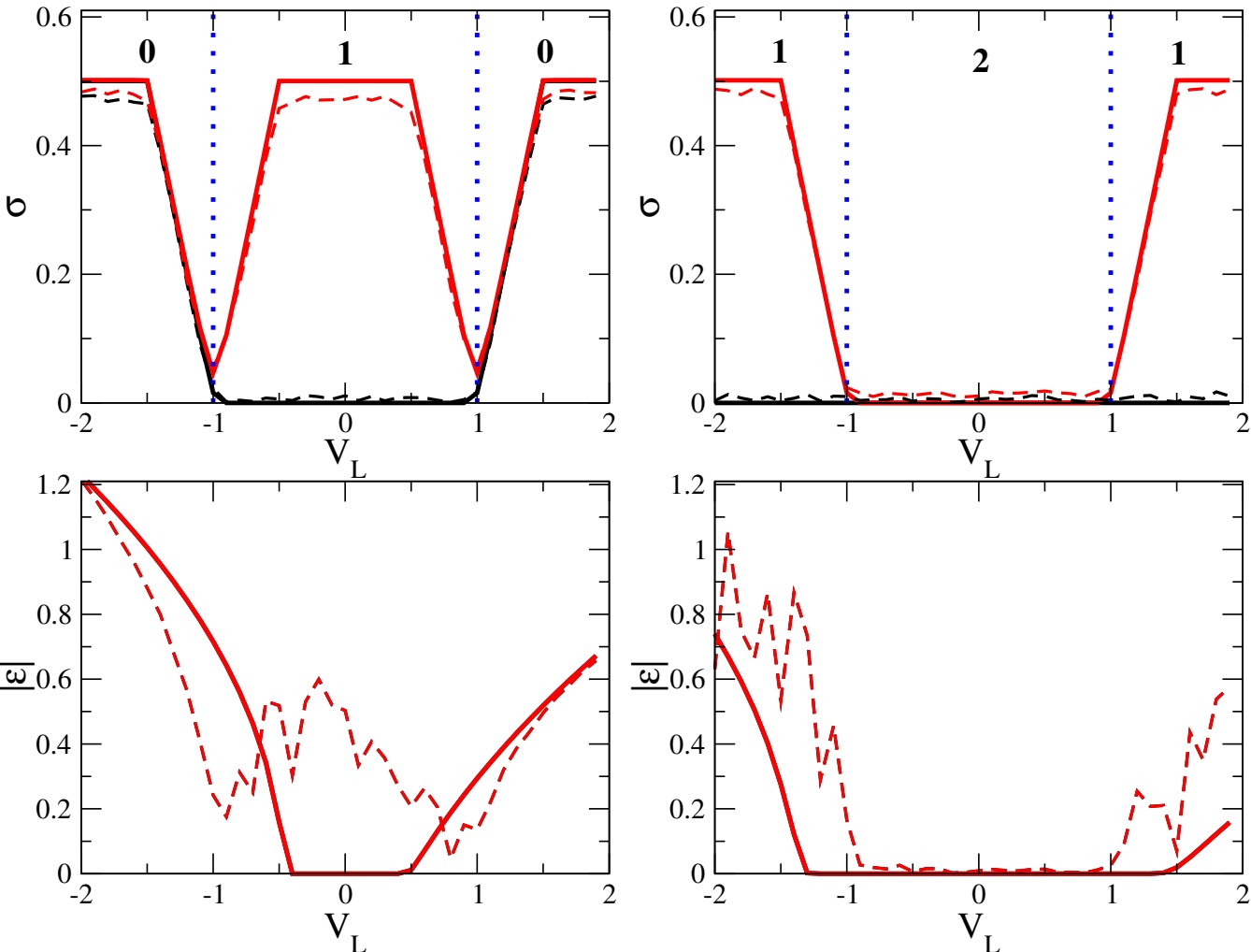

FIG. 8. The smallest two singular values (top row) and the smallest two eigenvalues (bottom row) for $W_L = 0.5$ (left column) and $W_L = 1.5$ (right column) with $V_R = W_R = 1$ for a system with $N = 200$. The two eigenvalues are always degenerate. The solid lines are for the unperturbed system, the dashed lines for a system where a random matrix with matrix elements $|a_{ij}| < 0.02$ has been added. The numbers in the top row indicate the sum of the winding numbers $|\nu_1| + |\nu_2|$.

as we have done above. Here we want to show explicitly how the exact edge states in the semi-infinite limit arise and that they become 'hidden' if the system is finite.

For a semi-infinite chain, we can find zero modes by looking for solutions of $Hv = 0$ and $\tilde{H}w = 0$. Here $\tilde{H} = H(h_j \to h_{-j})$ is the reflected Hamiltonian with Fourier components $h_j$. The reason that we have to consider also the reflected Hamiltonian is that we are thinking about a system which extends up to infinity either from an edge to the left or to the right.

The condition $Hv = 0$ leads to $v_2 = 0$ and $W_R v_{2j} + V_L v_{2j+2} = 0$ implying that all even vector components are zero, $v_{2j} = 0$. For the odd vector components we find the recurrence relation $V_R v_{2j-1} + W_L v_{2j+1} = 0$ with the solution $v_{2j+1} = \left(-\frac{V_R}{W_L}\right)^j v_1$. In addition, we have to demand that the solution is normalizable, $||v||^2 = |v_1|^2 \sum_{j=0}^{\infty} \left(-\frac{V_R}{W_L}\right)^{2j} = 1$. This is possible if

$|W_L| > |V_R|$. In this case, the semi-infinite chain has a zero energy edge mode $v$ exponentially localized at the left edge with vector components

$$v_{2j+1} = \sqrt{1 - \left(\frac{V_R}{W_L}\right)^2} \left(-\frac{V_R}{W_L}\right)^j, \quad v_{2j} = 0. \quad \text{(V.12)}$$

Note that the only solution of $Hv = 0$ for a finite system is the trivial vector $v = 0$.

Similarly, we can consider zero modes of the reflected Hamiltonian, $\tilde{H}w = 0$. In this case we find that an exponentially localized zero mode exists if $|W_R| > |V_L|$ and has vector components

$$w_{2j+1} = 0, \quad w_{2j+2} = \sqrt{1 - \left(\frac{V_L}{W_R}\right)^2} \left(-\frac{V_L}{W_R}\right)^j. \quad \text{(V.13)}$$

Again, for a finite system the only solution to $\tilde{H}w = 0$ is $w = 0$. The non-trivial edge mode with energy zero

only exists in the thermodynamic limit. We note that we find the exact same separation of parameters as for the singular values: the first zero mode depends only on $\{V_R, W_L\}$ and the second zero mode only on $\{V_L, W_R\}$. For a finite system, we can truncate the vectors $v, w$. They are then no longer eigenvectors, $Hv \neq 0$, but they do get mapped exponentially close to zero, i.e. $||Hv|| \to 0$ for $N \to \infty$ and similarly for $w$. This is what we mean by hidden zero modes.

For the example considered in Fig. 8 with $V_R = W_R = 1$ this means that the first zero mode is stable for $-1 < V_L < 1$ and the second zero mode for $|W_L| > 1$. This is fully consistent with the singular values as well as with the winding numbers as is expected based on the K-splitting theorem. We thus do have explicitly confirmed the bulk-boundary correspondence in this case.

To check that the singular values with $\lim_{N \to \infty} \sigma = 0$ are topologically protected while zero eigenvalues are, in general, not protected for a finite non-Hermitian system, we also show in Fig. 8 both quantities when calculated for a finite Hamiltonian $H$ ($N = 200$) with a random complex matrix $A$ with elements $|a_{ij}| \leq 0.02$ added. The results show that the singular values are stable and thus indeed topologically protected while the eigenvalues are, in general, not protected. More specifically, we find that for the case $W_L = 0.5$ shown in the lower left panel there are two degenerate zero eigenvalues in the unperturbed system for $-0.4 \lesssim V_L \lesssim 0.5$. However, the small perturbation immediately moves them away from zero energy. They are not protected.

The situation is slightly more complicated in the case $W_L = 1.5$ shown in the lower right panel of Fig. 8. In the unperturbed case, there are two degenerate zero energy eigenvalues for $-1.3 \lesssim V_L \lesssim 1.5$. Interestingly, adding a perturbation does not completely remove them from zero energy. Instead, they are stable for $-1 < V_L < 1$. This can be understood as follows: in this regime, the non-Hermitian Hamiltonian with $V_R = W_R = 1$ and $W_L = 1.5$ can be adiabatically connected to a chiral Hermitian Hamiltonian without closing the gap. This can be achieved, for example, by $W_R \to 1.5$ and then $V_R \to V_L$. For the obtained Hermitian Hamiltonian one can calculate the winding number of the upper block which turns out to be $I = 1$. The standard bulk-boundary correspondence then predicts the existence of two protected zero-energy edge modes. Since the gap never closes and the sub-lattice symmetry remains intact, these zero modes survive in the non-Hermitian case. However, the full topology of the non-Hermitian model also in this case is only captured by the singular values, not by the eigenvalues.

## VI.  ENTANGLEMENT ENTROPY

In this section we investigate the entanglement entropy for the non-Hermitian model (II.4) with periodic boundary conditions. We use the reduced density matrix ap-

proach for free fermions [52, 53] which has been extended to the non-Hermitian realm [26–28, 54, 55] using the generalized density matrix $\rho = |r\rangle\langle\ell|$ where $\langle\ell|$ and $|r\rangle$ are bi-orthogonal left and right eigenstates of (II.4). The key quantity is the two-point correlation matrix with elements

$$C_{ij} = \langle\ell|c_i^\dagger c_j|r\rangle = \text{tr}\left(\rho c_i^\dagger c_j\right) \qquad \text{(VI.1)}$$

Restricting the indices to a given subsystem $\mathcal{A}$, the von Neumann entanglement entropy is given by,

$$S_{\mathcal{A}} = -\sum_j \nu_j \log\nu_j + (1 - \nu_j)\log(1 - \nu_j) \qquad \text{(VI.2)}$$

where $\nu_j$ are the eigenvalues of the correlation matrix (VI.1) restricted to $\mathcal{A}$. In this paper we only consider $\mathcal{A} = \{1, \ldots, N/2\}$, that is, we cut the system in half. For the model considered in this paper, in contrast to Hermitian systems, the eigenvalues $\nu_j$ and $S_{\mathcal{A}}$ can be complex. Nevertheless, we will indiscriminately call $S_{\mathcal{A}}$ the entanglement entropy. In addition, we choose the branch cut of the logarithm to be on the negative real axis.

Also, in non-Hermitian systems, the notion of a ground state is not well defined since the spectrum is complex in general. We have to distinguish two cases. If there is a line gap, then we fill the states on one side of the line gap so that there is an excitation gap. If there is no line gap then constructing a ground state is somewhat arbitrary but we will describe in each case what the chosen ground state is.

To evaluate (VI.1), we first organize the eigenvalues in the desired ordering,

$$D = \text{diag}(\epsilon(k_{\sigma(1)}), \ldots, \epsilon(k_{\sigma(N)})) \qquad \text{(VI.3)}$$

where $\epsilon(k_{\sigma(j)})$ denotes one of the $N$ possible quasienergies. Accordingly, define the column vectors,

$$\vec{r}(k_{\sigma(j)}) = \sqrt{\frac{2}{N}}\begin{pmatrix} r_1(k_{\sigma(j)}) \\ r_2(k_{\sigma(j)}) \\ \vdots \\ r_N(k_{\sigma(j)}) \end{pmatrix}, \qquad \text{(VI.4)}$$

$$\vec{\ell}(k_{\sigma(j)}) = \sqrt{\frac{2}{N}}\begin{pmatrix} \ell_1(k_{\sigma(j)}) \\ \ell_2(k_{\sigma(j)}) \\ \vdots \\ \ell_N(k_{\sigma(j)}) \end{pmatrix}, \qquad \text{(VI.5)}$$

with components given by (III.1,III.2), which are respectively right and left eigenvectors of $\mathcal{T}_N$ with eigenvalue $\epsilon(k_{\sigma(j)})$. Introduce the matrices

$$R = \left(\vec{r}(k_{\sigma(1)}), \ldots, \vec{r}(k_{\sigma(N)})\right), \qquad \text{(VI.6)}$$

$$L = \left(\vec{\ell}(k_{\sigma(1)}), \ldots, \vec{\ell}(k_{\sigma(N)})\right). \qquad \text{(VI.7)}$$

It follows that,

$$\mathcal{T}_N R = RD, \quad L^T \mathcal{T}_N = DL^T, \quad RL^T = L^T R = \mathbb{1}. \tag{VI.8}$$

The diagonalized hopping matrix is then given by

$$\mathcal{T}_N = \boldsymbol{\eta}^\dagger D \boldsymbol{\eta}, \quad \boldsymbol{\eta} = L^T \boldsymbol{c}, \quad \boldsymbol{\eta}^\dagger = \boldsymbol{c}^\dagger R \tag{VI.9}$$

implying the algebra (II.2) also for the $\boldsymbol{\eta}$. We have,

$$c_j = \sum_{m=1}^N R_{jm} \eta_m, \quad c_i^\dagger = \sum_{n=1}^N L_{in} \eta_n^\dagger. \tag{VI.10}$$

Finally, define the states,

$$\langle \ell | = \langle 0 | \prod_{a \in occ.} \eta_a, \quad |r\rangle = \prod_{a \in occ.} \eta_a^\dagger |0\rangle, \tag{VI.11}$$

and compute,

$$C_{ij} = \langle \ell | c_i^\dagger c_j | r \rangle = \sum_{a \in occ.} L_{ia} R_{ja}. \tag{VI.12}$$

We now investigate the spectrum of the correlation matrix (VI.12) and the associated entanglement entropy for indices in (VI.12) restricted to $i, j \in \{1, \dots, N/2\}$. We perform numerical calculations for large lattices, in which case we only keep eigenvalues of the correlation matrix that satisfy $|\nu_j| > 10^{-50}$ and $|1 - \nu_j| > 10^{-50}$. Additionally, we set to zero real or imaginary parts of $\nu_j$ and $1 - \nu_j$ that are individually smaller than $10^{-50}$. This

is particular important in cases where the eigenvalues of the correlation matrix are negative with a tiny imaginary part, whose sign can numerically oscillate around the branch cut. Before considering numerics, as a warm-up, we study the trivial 2-cell model ($N = 4$) analytically. Both zero field $u = 0$ and $\mathcal{PT}$-symmetric cases are considered.

## A. Sub-lattice symmetric case

For the zero field case, setting $V_R = W_R = 1$, let us consider transitions between the phases $(\nu_1, \nu_2) = (1, -1)$ (with $W_L > 1$) and $(0, -1)$, between $(1, 0)$ and $(0, 0)$ and between $(1, -1)$ (with $W_L < -1$) and $(0, -1)$, recall the phase diagram shown in the left panel of Fig. 2. For concreteness, let us consider only $V_L \geq 0$. Recall that the spectrum of $\mathcal{T}_N$ in the phases $(1, -1)$ and $(0, 0)$ is in general composed of two lobes in the complex plane either separated by the imaginary axis or real axis, therefore being characterized by $I_1 = 0$. On the other hand, the spectrum in the phases $(1, 0)$ and $(0, -1)$ traces a closed path around $E_B = 0$, therefore being characterized by $I_1 \neq 0$.

We start with the 2-cell model ($N = 4$). For this lattice size, there is no lobe in the spectrum, and the eigenvalues are either purely real or purely imaginary. We fill the states with energy $-\sqrt{(V_L \mp 1)(1 \mp W_L)}$ and leave empty the states with energy $\sqrt{(V_L \mp 1)(1 \mp W_L)}$. The associated correlation matrix is given by,

$$C = \begin{pmatrix} \frac{1}{2} & -\frac{\sqrt{(V_L-1)(1-W_L)}}{4(V_L-1)} - \frac{\sqrt{(V_L+1)(W_L+1)}}{4(V_L+1)} \\ -\frac{\sqrt{(V_L-1)(1-W_L)}}{4(1-W_L)} - \frac{\sqrt{(V_L+1)(W_L+1)}}{4(W_L+1)} & \frac{1}{2} \end{pmatrix}, \tag{VI.13}$$

which is non-Hermitian and can have complex eigenvalues. Indeed, we plot both real and imaginary parts of the two eigenvalues of (VI.13) in Fig. 9 for some fixed values of $W_L$ as function of $V_L$.

We observe that within the phases characterized by $(1, -1)$ the eigenvalues form conjugate pairs of the form $1/2 \pm i\alpha$, see panels (a) and (c) in Fig. 9 for the fixed values $W_L = \pm 1.5$. The same form of eigenvalues is observed for other values of $|W_L| > 1$. Thus, in this phase, the entanglement entropy is real and given by,

$$S_2 = -\log\left(\frac{1}{4} + \alpha^2\right) + 4\alpha \arctan(2\alpha) \tag{VI.14}$$

where $\alpha$ depends on $V_L$ and $W_L$. We also note that $\alpha \to \infty$ as the transition point $V_L = 1$ is approached and, as a consequence, $S_2$ diverges at $V_L = 1$. On the other hand, deep inside the phase, taking $V_L = 0$ and $W_L \to \infty$, we have $\alpha \to 0$ and therefore $S_2 \to 2\log 2$.

Increasing $V_L$ and crossing the transition point $V_L = 1$,

we observe that the eigenvalues of the correlation matrix in the phase $(0, -1)$ are complex, with the form $\beta_{1,2} \pm i\alpha$. That is, the eigenvalues have imaginary parts with opposite signs but with different real parts $\beta_2 = 1 - \beta_1$. This leads to complex values for the entanglement entropy. However, for large $|W_L|$ the imaginary part tends to disappear when $V_L$ increases. In the limit $V_L \to \infty$ and $|W_L| \to \infty$, the exact eigenvalues of the correlation matrix are $1/2 \pm \sqrt{2}/4$, producing a real entropy $S_2 \approx 0.832991$.

In the phase $(1, 0)$, the eigenvalues once more have the form $\beta_{1,2} \pm i\alpha$ with $\beta_2 = 1 - \beta_1$, and the entanglement entropy is complex (except if $V_L = W_L$), see panel (b) in Fig. 9 for the fixed value $W_L = 0.1$. Moving to the phase $(0, 0)$, we observe that the eigenvalues of $C$ are real, however, they are greater than unity or smaller then zero, and therefore also produce a complex entanglement entropy. Only in the limit $V_L \to \infty$ and $W_L \to 0$, do the eigenvalues of the correlation matrix tend to 1 and 0,

thus producing a zero entanglement entropy.

Now, we consider larger lattice sizes using numerical diagonalization. We observe a variety of possible types of eigenvalues in the correlation spectrum, similar to the trivial 2-cell model. In some phases, we may find eigenvalues with $\text{Re}(\nu_j) > 1$ as well as $\text{Im}(\nu_j) \neq 0$. Also, we note that near the transition points the behaviour of the correlation matrix eigenvalues is more intricate. Let us analyze each phase in details. Results are collected in Fig. 10 for a lattice composed of $N = 120$ sites. In the insets of Fig. 10, we show the typical form of the complex spectrum indicating the filling adopted, as well as the real part of the eigenvalues of the correlation matrix, considering a smaller lattice with $N = 52$ for better visualization of the results.

Starting with the phases $(-1, 1)$, see panels (a) and (c) in Fig. 10 for the fixed values $W_L = \pm 1.5$, we observe that most of the eigenvalues of the correlation matrix are either close to 1 or 0. Some of the eigenvalues have the property $\nu_j > 1$ or $\nu_j < 0$, and there are two eigenvalues sitting at $1/2 \pm i\alpha$. As the lattice size increases, we observe that those eigenvalues with $\nu_j > 1$ or $\nu_j < 0$ tend to unity or zero, while $\alpha$ decreases. For instance, for $N = 52$, the lattice size considered in the insets, the highest eigenvalue is $\nu_{\max} \approx 1.000000000290214$, the lowest $\nu_{\min} = 1 - \nu_{\max}$ and $\alpha \approx 4.275663 \times 10^{-4}$, while for $N = 120$ the highest eigenvalue for $W_L = 1.5$ and $V_L = 0.5$ is $\nu_{\max} \approx 1.0000000000000000265940$, the lowest $\nu_{\min} = 1 - \nu_{\max}$ and $\alpha \approx 2.955440 \times 10^{-7}$. Therefore, the entanglement entropy comes essentially from the conjugate eigenvalues $1/2 \pm i\alpha$, and its value is also given by (VI.14). For a fixed length of the chain, but going deeper into the phase by increasing $|W_L|$, we also note that $\alpha$ decreases. Then, in the limit $\alpha \to 0$, the entanglement entropy has the finite value $S_{N/2} \to 2 \log 2$ deep inside the phase. Similar considerations can be made for the phase $(1, -1)$ with $W_L < -1$; note however that a different filling must be used in this case.

Moving to the phases $(0, -1)$, see once more panels (a) and (c) in Fig. 10, we note that all eigenvalues of the correlation matrix satisfy $0 < \text{Re}(\nu_j) < 1$, with most eigenvalues being either close to unity or zero. Some intermediate eigenvalues appear, and these carry a small imaginary part. They produce a finite entanglement entropy, also complex, but with a small imaginary part. However, the number of intermediate states actually grows with the length of the chain. This implies a scaling of the entanglement entropy, see Fig. 11, where we fix $W_L = 1.5$

and some values of $V_L$ deep inside the phase. We observe a scaling of the form $\text{Re}(S_N) \sim (c/3) \log(N)$ with $c = 1$, as well as a decay $\text{Im}(S_N) \sim \exp(-\log(N))$ of the imaginary part. It follows that, remarkably, this phase may be described by a CFT with central charge $c = 1$, even though it has a point gapped spectrum. Similar results can be obtained for $W_L < -1$ in the same phase $(0, -1)$.

The phase $(1, 0)$, see panel (b) in Fig. 10, is similar to the phases $(0, -1)$. The scaling of the entanglement entropy for this case is shown in Fig. 12. Also in this phase the scaling of the entanglement entropy is consistent with a central charge $c = 1$.

The last phase to be discussed in this subsection is the phase $(0, 0)$. Here, we observe eigenvalues of the correlation matrix with $\text{Re}(\nu_j) > 1$ and also some eigenvalues with a small imaginary part. We note that, for fixed parameters $W_L$ and $V_L$ in this phase, the imaginary part of the entanglement entropy decreases with the length of the chain, while the real part slightly increases and saturates. However, as we go deeper into this phase by increasing $V_L$, the entanglement entropy tends to disappear. This phase is therefore trivial.

In summary, for the zero field case and in the thermodynamic limit, the entanglement entropy is real deep inside the quantum phases, and discontinuous near the transition points. The gapped non-trivial phases have two protected eigenvalues at $1/2$, contributing $S_N = 2 \log 2$ to the entanglement entropy.

## B. $\mathcal{PT}$-symmetric case

Let us now consider the $\mathcal{PT}$ symmetric case with PBC, setting $u = 1$. For simplicity, we will restrict ourselves to the quadrant with $V \geq 0$ and $W \geq 0$ in the phase diagram shown in the right panel of Fig. 2. Then, four regions must be analyzed. The region $|W - V| < 1 < W + V$ is the exceptional phase, characterized by a large number of exceptional points (recall Sec. III B). In this phase, the $\mathcal{PT}$-symmetry is broken. The regions $W - V < -1$ and $W - V > 1$ are adiabatically connected to the standard SSH model via $u \to 0$. Then, the former is called the $\mathcal{PT}$-unbroken trivial phase and the latter the $\mathcal{PT}$-unbroken topological phase. In the phase $W + V < 1$ the spectrum is purely imaginary, and we name it the complex phase.

As before, we start our analysis by the simple 2-cell model ($N = 4$), which has the following correlation matrix,

$$C = \frac{1}{4} \begin{pmatrix} \frac{(V-W)^2}{(V-W)^2 - 1 + i\sqrt{(V-W)^2-1}} + \frac{(V+W)^2}{(V+W)^2 - 1 + i\sqrt{(V+W)^2-1}} & \frac{W-V}{\sqrt{(V-W)^2-1}} - \frac{V+W}{\sqrt{(V+W)^2-1}} \\ \frac{W-V}{\sqrt{(V-W)^2-1}} - \frac{V+W}{\sqrt{(V+W)^2-1}} & \frac{(V-W)^2}{(V-W)^2 - i\sqrt{(V-W)^2-1}-1} + \frac{(V+W)^2}{(V+W)^2 - i\sqrt{(V+W)^2-1}-1} \end{pmatrix}$$

associated with the filled states $-\sqrt{(V \mp W)^2 - 1}$ and empty states $\sqrt{(V \mp W)^2 - 1}$. We plot in Fig. 13 the

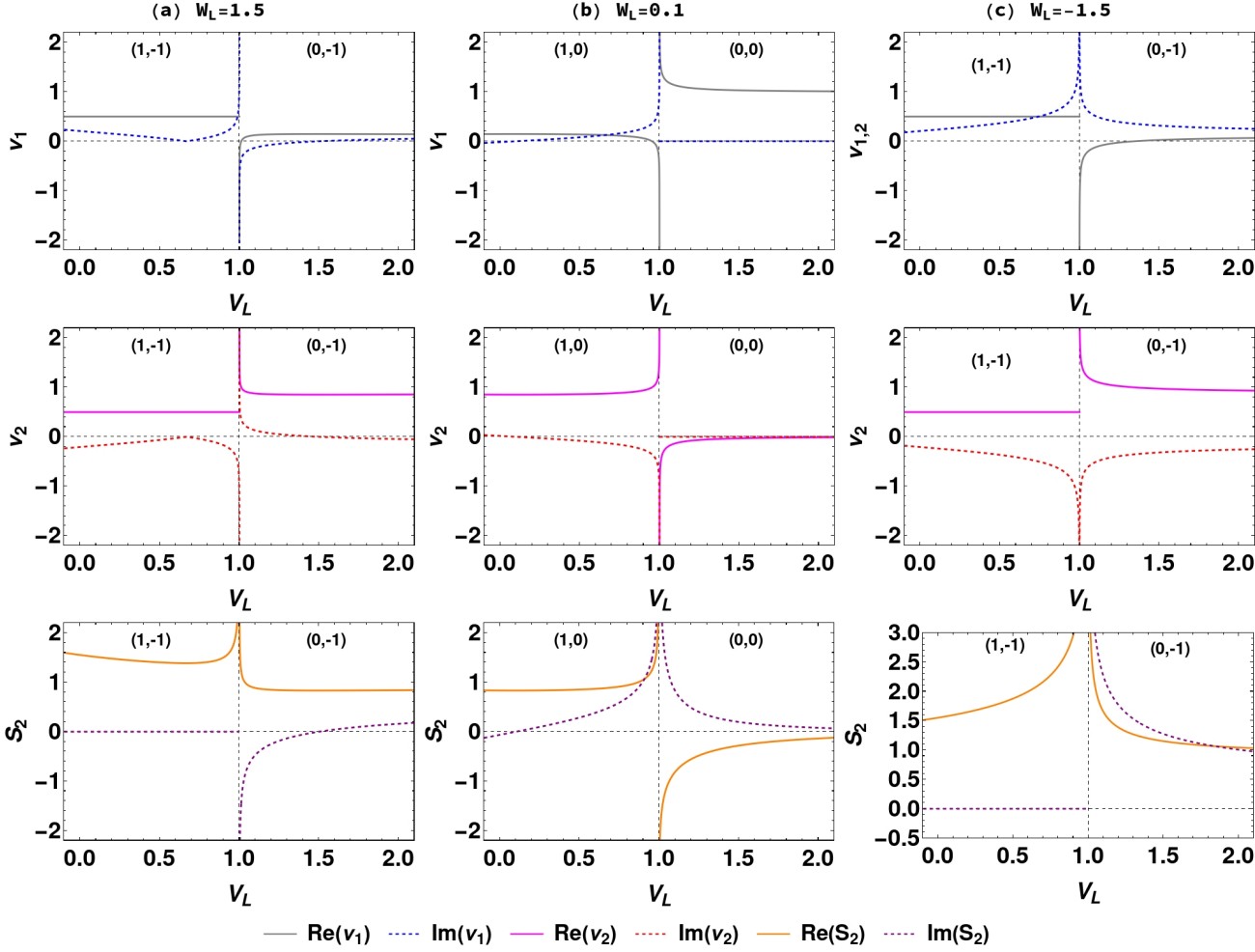

FIG. 9. Eigenvalues of the correlation matrix (top and middle panels) and entanglement entropy (bottom panels) for the trivial two-cell model ($N = 4$) with zero field $u = 0$, $W_R = V_R = 1$ and some fixed values of $W_L$ indicated in the labels (a)-(c) as a function of $V_L$.

eigenvalues of $C$ and the associated entanglement entropy for some fixed values of $V$ as a function of $W$, covering all the four regions of the phase diagram.

For $V = 0$ and $0 < W < 1$ that is, within the complex phase, see panel (a) in Fig. 13, we can observe that the eigenvalues of the correlation matrix are real with $\nu_j > 1$ or $\nu_j < 0$ until the phase transition point $W = 1$ is reached. As a consequence, the entanglement entropy is complex in this phase. Crossing the transition point ($W = 1$) towards the $\mathcal{PT}$-unbroken topological phase, we note that the eigenvalues have the form $1/2 \pm i\alpha$, and the entanglement entropy is given by (VI.14). In the limit $W \to \infty$, $\alpha$ becomes negligible and the entanglement entropy tends to $S_2 \to 2\log(2)$, indeed characteristic of a topological phase with winding $I = 1$.

Increasing $V$ to $V = 0.5$, see panel (b) in Fig. 13, the behaviour of the eigenvalues $\nu_j$ is the same as for $V = 0$ until the exceptional phase is reached at $W = 0.5$. Within the exceptional phase, the eigenvalues $\nu_j$ have the form $\beta_{1,2} \pm i\alpha$, that is, the eigenvalues have the

same imaginary part with opposite signs but different real parts related by $\beta_2 = 1 - \beta_1$. As a consequence, the entanglement entropy is complex in the exceptional phase. This phase ends at $W = 1.5$ from which point the $\mathcal{PT}$-unbroken topological phase is again reached being characterized by a real entanglement entropy which tends to $S_2 \to 2\log(2)$ deep inside the phase.

To conclude the analysis of the 2-cell model, we consider $V = 1.5$ such that the $\mathcal{PT}$-unbroken trivial phase is considered. This phase for $V = 1.5$ occurs when $0 < W < 0.5$, and we note that there the eigenvalues are real with $\nu_j > 1$ or $\nu_j < 0$, producing again a complex entanglement entropy except if $W = 0$. However, deep inside the phase ($V \to \infty$), the eigenvalues tend to 0 and 1, leading to a null entanglement, as expected from a trivial phase. For $0.5 < W < 2.5$ we encounter once more the exceptional phase and for $W > 2.5$ the $\mathcal{PT}$-unbroken topological phase.

Finally, we consider the $\mathcal{PT}$ case for large lattice sizes. Results for the correlation spectrum are collected in

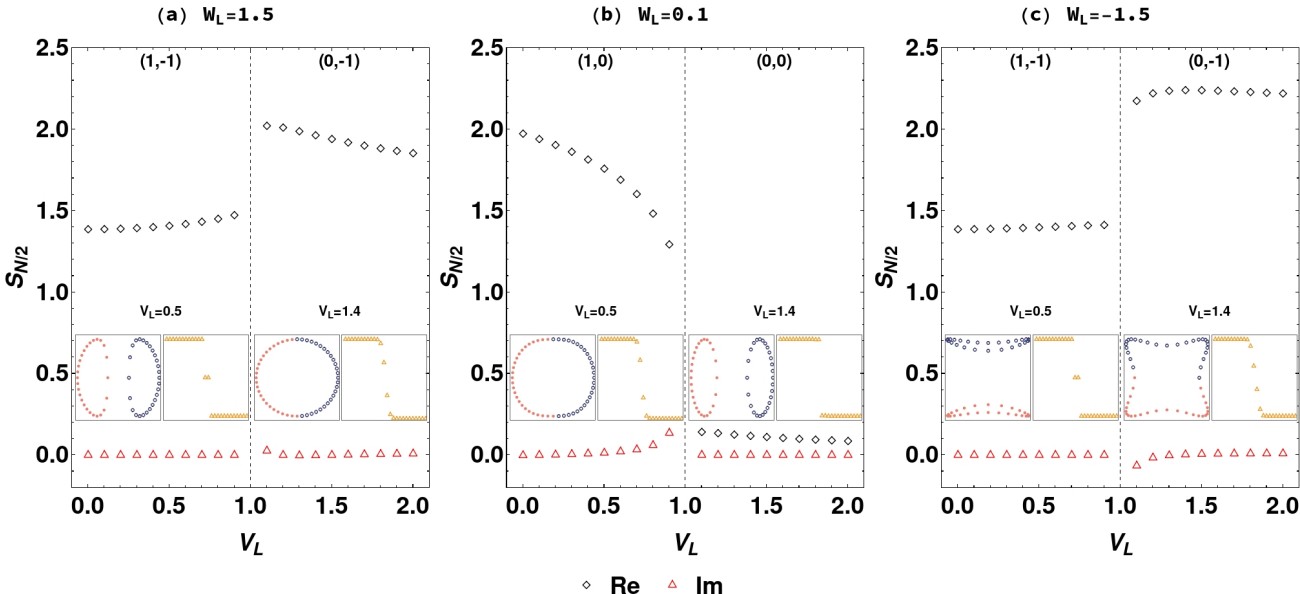

FIG. 10. Entanglement entropy for the model with sublattice symmetry (zero field $u = 0$) for $V_R = W_R = 1$ and some fixed values of $W_L$ indicated in the labels (a)–(c), as a function of $V_L$. The lattice size is $N = 120$. In the insets, the typical form of the complex spectrum is shown; the occupied states are indicated by filled circles while the unoccupied states by open circles. The typical eigenvalues of the correlation matrix are represented by empty up triangles (only the real part is shown since the complex part when present is small). For better visualization, in the insets we consider $N = 52$.

Fig. 14 (where $N = 52$) and for the entanglement entropy in Fig. 15 (where $N = 120$). The same four regions considered for the 2-cell model are analyzed.

First, let us consider $V = 0$ and vary $W$. For $0 < W < 1$, we note that the correlation matrix has real eigenvalues, most of them 0 or 1, but there is one eigenvalue $\nu_{\max} > 1$ and one eigenvalue $\nu_{\min} < 0$. Remarkably, these eigenvalues do not depend on the lattice size, and therefore correspond to the two eigenvalues of the 2-cell model. For instance, for $V = 0, W = 0.5$, the eigenvalues are $\nu_{\max} \approx 1.077350$ and $\nu_{\min} \approx -0.077350$, which produces a finite entanglement entropy with negative real part and positive imaginary part, precisely the same as the 2-cell model, recall panel (a) in Fig. 13. Crossing the transition point $W = 1$, we note that most of the eigenvalues of the correlation matrix are 0 or 1, but there are a couple of eigenvalues of the form $1/2 \pm i\alpha$, where $\alpha$ also does not depend on the lattice size. However, we note that for a fixed length $\alpha \to 0$ when $W \to \infty$. Therefore, like in some of the previously considered cases, the entanglement approaches $S_{N/2} \to 2\log(2)$ deep inside this phase.

Now we consider $V \neq 0$. Similarly to the 2-cell model, by fixing $0 < V < 1$, we can reach the complex phase, the exceptional phase and the $\mathcal{PT}$-unbroken topological phase by varying $W$. On the other hand, by fixing $V > 1$ and varying $W$ we can reach the $\mathcal{PT}$-unbroken trivial phase, the exceptional phase and the $\mathcal{PT}$-unbroken topological phase.

Let us consider once more $V = 0.5$. Within the complex phase ($0 < W < 0.5$ for $V = 0.5$), all the eigenvalues of the correlation matrix jump above unity or below zero, see panel (a) in Fig. 14 where the parameters $W = 0.45$ and $V = 0.5$ are used. This results in a complex value for $S_{N/2}$, see panel (a) in Fig. 15. We note that two eigenvalues stand out, being significantly greater than 1 or less than 0. All the eigenvalues of the correlation matrix in this phase seem to vary slowly with the size of the chain, actually tending to stabilize. For example, the difference between the largest eigenvalues for the lattices with $N = 52$ and with $N = 120$ is of the order of $10^{-9}$. Therefore, the entanglement entropy in this phase seems to be finite. However, it is difficult to verify this property since the model parameters in this phase lie in a finite interval. Also, the phase is surrounded by critical lines that drive the entanglement entropy to high values.

Moving to the $\mathcal{PT}$-unbroken topological phase, we note that the correlation spectrum is composed by eigenvalues close to 1 and 0, with some of them being slightly bigger than 1 or smaller then 0, as well as a conjugate pair of the form $1/2 \pm i\alpha$, see panel (b) in Fig. 14 where the parameters are set to $W = 1.7, V = 0.5$. As the lattice size increases, we note that the eigenvalues tend to 1 or 0, while $\alpha$ decreases slowly and seems to stabilize with the length of the chain. The associated entanglement entropy is real and finite. Going deep into the phase by increasing $W$, as before, $\alpha$ decreases and the entanglement tends to $S_{N/2} \to 2\log 2$, see both panels (a) and (b) in Fig. 15.

In the $\mathcal{PT}$-unbroken trivial phase, the correlation spectrum is also composed mostly of 0's and 1's, some of them slightly bigger than 1 or smaller than 0. Also, some eigenvalues carry an imaginary part in the form of conjugate

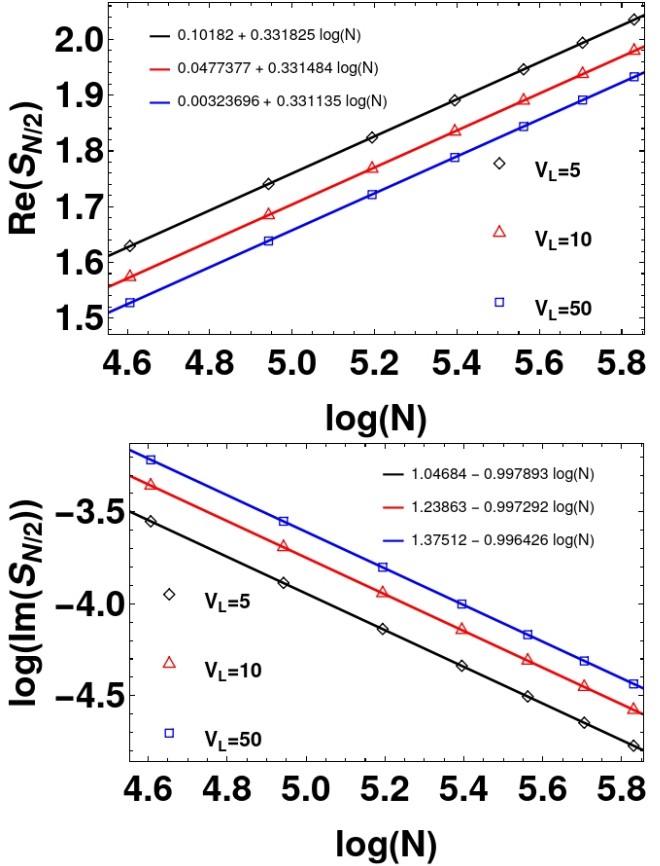

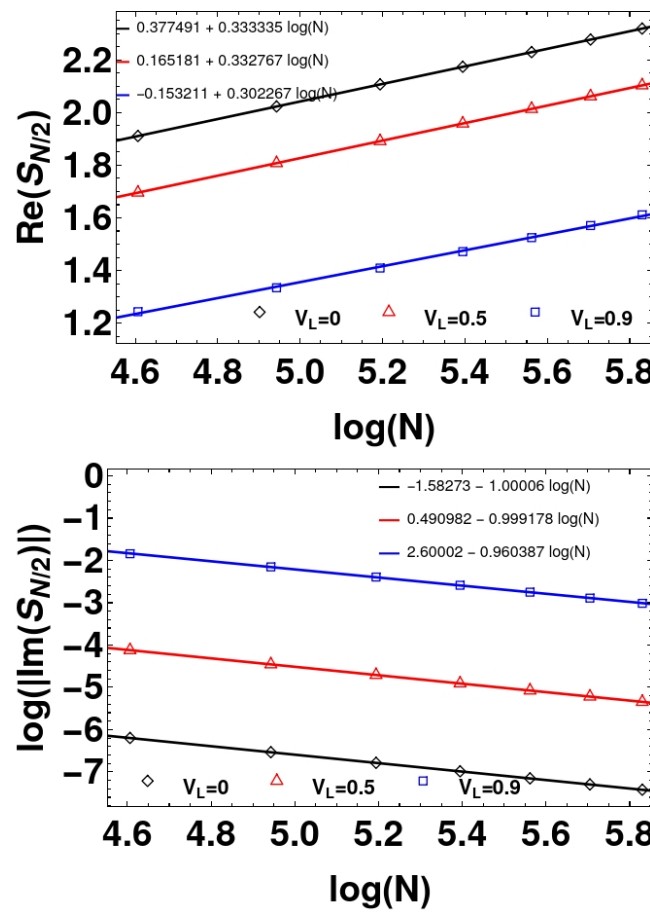

FIG. 11. Scaling of the entanglement entropy in the phase $(0, -1)$ for $W_L = 1.5$ and values of $V_L$ deep inside the phase.

FIG. 12. Scaling of the entanglement entropy in the phase $(1, 0)$ for $W_L = 0.1$ and some values of $V_L$ inside the phase. Note that for $V_L = 0$ the imaginary part is negative, so it is its absolute value that decreases with the length of the chain.

pairs, see panel (d) in Fig. 14 for the model parameters $W = 0.5$ and $V = 1.8$. Interestingly, similarly to previous cases, both real and imaginary part of the eigenvalues vary slowly with the length of the chain. Nevertheless, deep inside this phase (large $V$), the eigenvalues tend to 0 or 1 and produces zero entanglement, as expected from a trivial phase; see panel (b) in Fig. 15.

Let us briefly discuss the exceptional phase. In this phase we observe an intricate behaviour of the correlation spectrum, characterized by various complex eigenvalues, see panel (c) in Fig. 14 where the parameters are set to $V = 0.5$, $W = 1.2$. This behaviour is reflected in an intricate evolution of the entanglement entropy, see panels (a) and (b) in Fig. 15. Interestingly, the entanglement entropy detects the presence of the real exceptional points (indicated by dashed lines in Fig. 15). For clarity, we only pick a few values of $W$ between the exceptional points. Note that in the thermodynamic limit the entanglement entropy is ill defined in the exceptional phase, since it is filled with exceptional points. For finite systems, we expect a scaling of $S_{N/2}$. This is numerically challenging and goes beyond the scope of this paper.

In brief, the entanglement entropy has a rich behaviour in the $\mathcal{PT}$ symmetric model and is capable of detecting exceptional points and phase transitions. In the $\mathcal{PT}$-

unbroken topological phase there are two protected eigenvalues at $1/2$, contributing $2\log 2$ to the entanglement entropy.

## VII. CONCLUSION

In this paper we have analytically studied a paradigmatic one-dimensional, non-Hermitian lattice model. Our goal was to illustrate many of the unique features of non-Hermitian systems in a concrete model and to do so largely based on analytical instead of purely numerical results. We derived the eigensystems both for closed (periodic and anti-periodic) as well as open boundary conditions. Our analytical results show how the Bloch waves in the closed case change into localized eigenstates for open boundaries thus explicitly demonstrating the non-Hermitian skin effect. Another common feature of non-Hermitian systems are exceptional points where both eigenvalues and -vectors coalesce. In the considered model, we have shown that such points typically occur when the complex gap closes and that the loca-

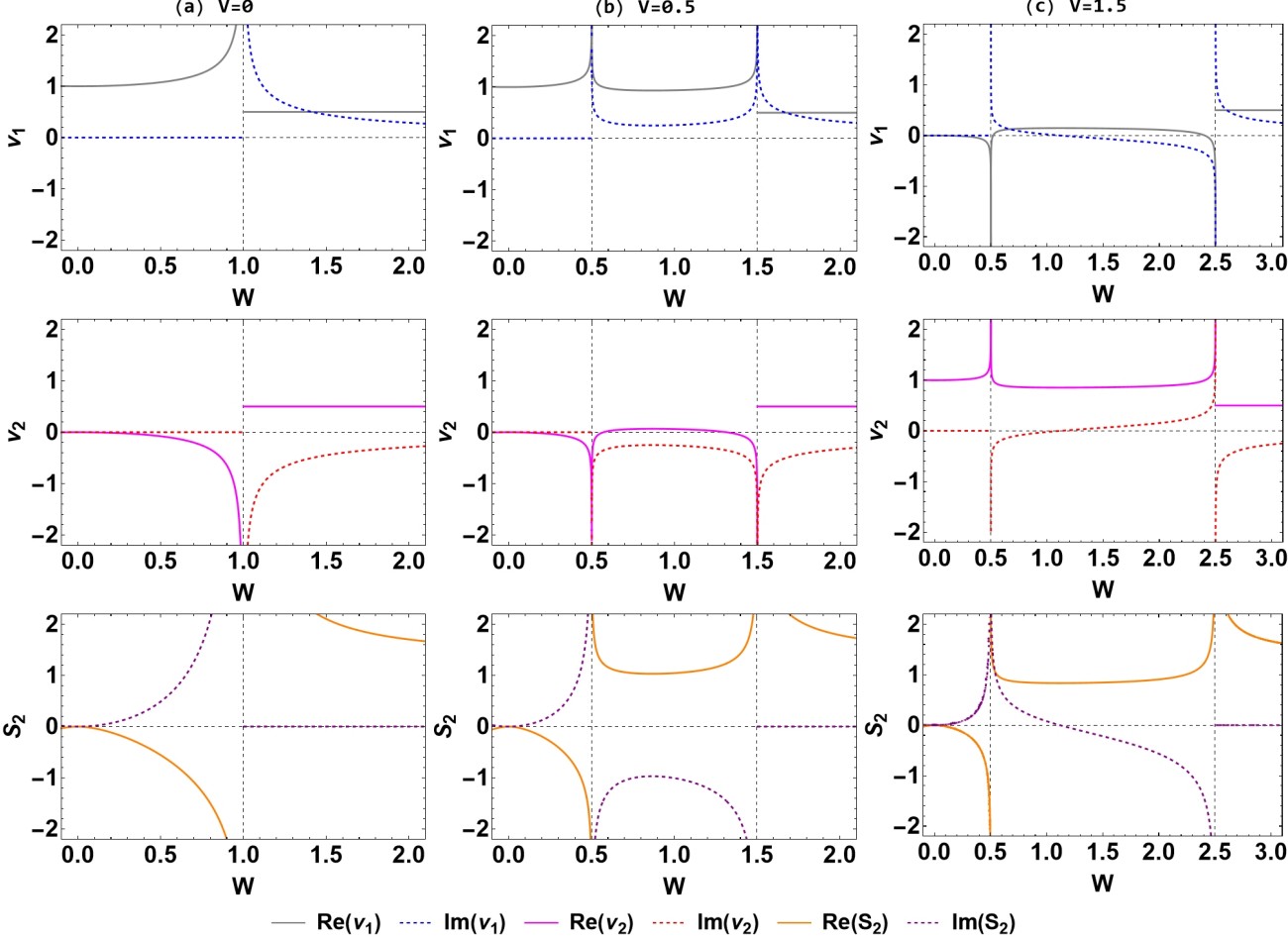

FIG. 13. Eigenvalues of the correlation matrix (top and middle panels) and entanglement entropy (bottom panels) for the trivial $\mathcal{PT}$-symmetric two-cell model ($N = 4$) with $u = 1$, some fixed values of $V$ indicated in the labels (a)-(c) as a function of $W$.

tion of exceptional points, in general, differs between the closed and the open case. Surprisingly, we found in the $\mathcal{PT}$-symmetric phase that exceptional points can even become dense in the thermodynamic limit, forming an entire "exceptional phase".

One of the central results of this work is an explicit demonstration of a proper bulk-boundary correspondence for a non-Hermitian system. This point has lead to some confusion in the literature. Here we have shown that the winding numbers for the periodic system determine the number of protected singular values which go to zero in the thermodynamic limit. This number also corresponds to the number of exact zero-energy edge modes for a semi-infinite chain. However, the eigenspectrum of the finite chain does, in general, not converge to that of the semi-infinite one. Thus, no bulk-boundary correspondence between winding numbers and the eigenspectrum of a finite system exists, in contrast to Hermitian systems. The exact zero-energy edge states in the semi-infinite non-Hermitian case are not exact eigenstates for a finite system. They become hidden which

means that they correspond to metastable states which are, however, very long-lived and thus highly relevant for our understanding of non-Hermitian systems; see also Ref. [25] where this issue was discussed more generally. It is also important to note that even if zero-energy eigenstates for a finite system do exist, they are typically not topologically protected. For the studied model we demonstrated this explicitly by adding a small perturbation to the Hamiltonian and showing that the eigenenergies shift while the topologically protected singular values are stable.

Another aspect of topological protection in non-Hermitian systems which we explored here is the entanglement spectrum. For cases with a line gap we could, in particular, define a half-filled system where occupied and unoccupied bands are separated by a gap. Since the studied system is Gaussian, the entanglement spectrum is fully determined by the spectrum of the single-particle correlation matrix. We found that phase transitions and exceptional points are indicated by discontinuities in the entanglement entropy. Furthermore, we demonstrated

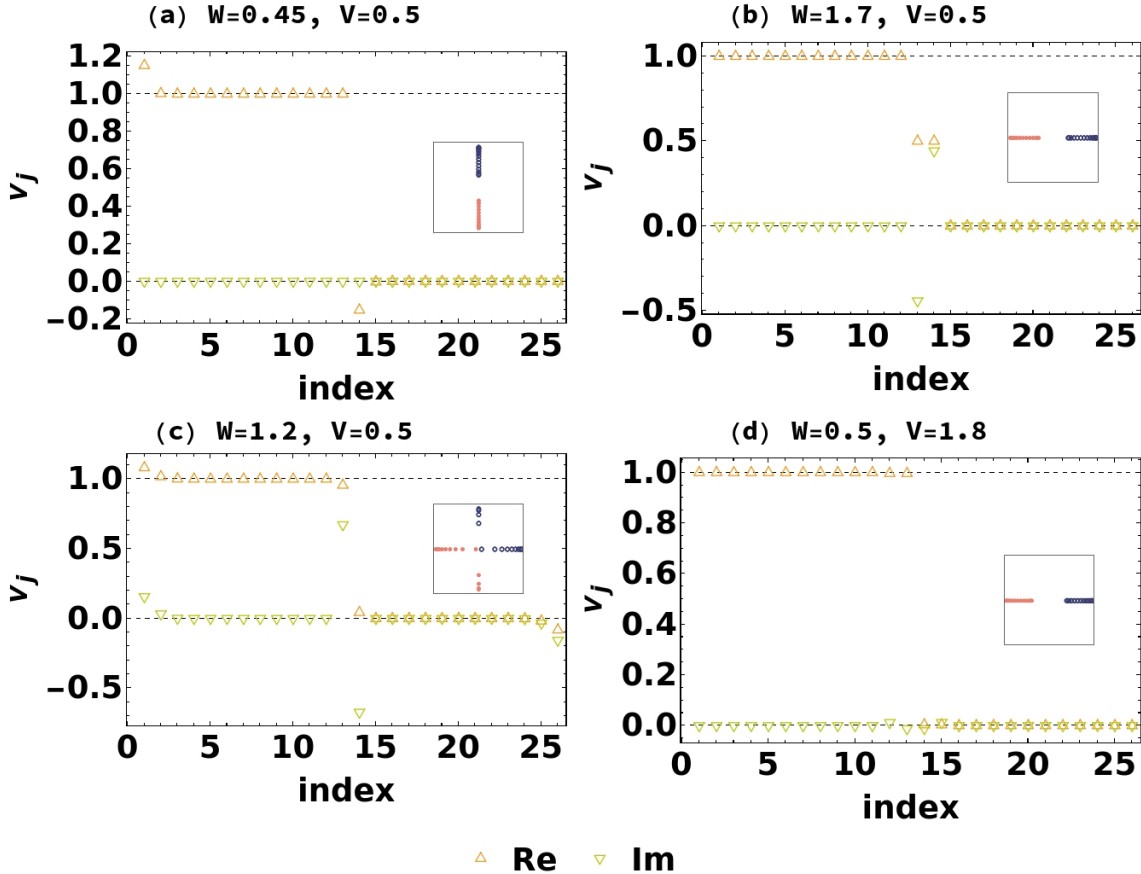

FIG. 14. Eigenvalues of the correlation matrix for the $\mathcal{PT}$ symmetric case with PBC and $u = 1$ for some fixed values of $W$ and $V$ indicated in the labels (a)-(d). The lattice size is $N = 52$. In the insets, the complex spectrum is shown; occupied states are indicated by orange filled circles and unoccupied states by blue open circles.

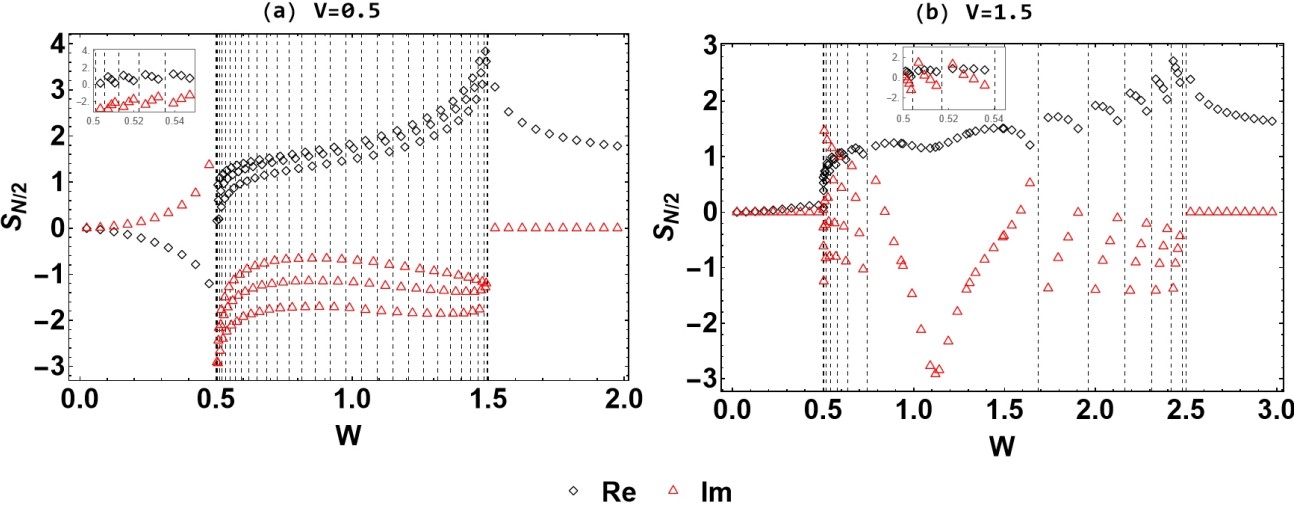

FIG. 15. Entanglement entropy for the $\mathcal{PT}$ symmetric case with PBC and $u = 1$ as a function of $W_L = W_R = W$ for two fixed values of $V_L = V_R = V$ as indicated in the labels. The lattice size is $N = 120$. The dashed grid lines indicate the position of real exceptional points. In the insets, we zoom into regions where the exceptional points are very close to each other.

that in topologically non-trivial phases there are protected eigenvalues in the entanglement spectrum which lead to lower, non-trivial entanglement bounds.

Non-trivial topology in the studied non-Hermitian system thus leads to two observable and experimentally relevant phenomena: (1) Extremely long-lived stable zero energy states, and (2) non-trivial entanglement in systems with a gap between occupied and unoccupied energy bands.

## Appendix A: Some details for the open boundary case

To make this paper self-contained, we review some details of the diagonalization of $\mathcal{T}_N$ (II.4) for open boundary conditions, that is, $\gamma = 0$, following [44, 45].

The approach in [44] is based on recurrence relations satisfied by the characteristic polynomial,

$$P_N(z) = \det_N (\mathcal{T}_N - z). \qquad (A.1)$$

The roots $P_N(z) = 0$ are the quasienergies of the Hamiltonian (II.1). The characteristic polynomial (A.1) for $\gamma = 0$ is special as it satisfies the recurrence relations

$$
\begin{aligned}
P_{2j}(z) &= (-\mathrm{i}u - z)P_{2j-1}(z) - V_L V_R P_{2j-2}(z), \\
P_{2j-1}(z) &= (\mathrm{i}u - z)P_{2j-2}(z) - W_L W_R P_{2j-3}(z)
\end{aligned} \qquad (A.2)
$$

depending on the parity of $N$. Note that one has an additional recurrence relation for the characteristic polynomial, namely

$$
\begin{aligned}
P_j(z) &= \left((z + \mathrm{i}u)(z - \mathrm{i}u) - (V_L V_R + W_L W_R)\right) P_{j-2}(z) \\
&\quad - V_L V_R W_L W_R P_{j-4}(z),
\end{aligned} \qquad (A.3)
$$

where appropriate initial conditions must be chosen.

On the other hand, following [45], one notes that equations (II.6,II.7) can be iterated to the same fourth order difference equation,

$$
\begin{aligned}
r_{j+2} &+ \frac{V_R W_R}{V_L W_L} r_{j-2} \\
&= \left(-\frac{(\mathrm{i}u + \epsilon)(\mathrm{i}u - \epsilon) + (V_L V_R + W_L W_R)}{V_L W_L}\right) r_j.
\end{aligned} \qquad (A.4)
$$

The difference equation (A.4) can be compared to (A.3) leading to the solution (IV.10).

In addition, following [45], introduce

$$T_n(\alpha, \beta) = \frac{\alpha^{n+1} - \beta^{n+1}}{\alpha - \beta} \qquad (A.5)$$

and the sequence

$$a_{n+2} - (\alpha + \beta)a_{n+1} + \alpha\beta a_n = 0. \qquad (A.6)$$

Then, it follows that

$$a_n = T_{n-2}(\alpha, \beta)a_2 - \alpha\beta T_{n-3}(\alpha, \beta)a_1 \qquad (A.7)$$

is the solution of the difference equation (A.6). Now choosing

$$\alpha + \beta = \frac{(\mathrm{i}u - \epsilon)(-\mathrm{i}u - \epsilon) - (V_L V_R + W_L W_R)}{V_L W_L} \qquad (A.8)$$

$$\alpha\beta = \frac{V_R W_R}{V_L W_L}, \qquad (A.9)$$

it follows that

$$r_{2j} = T_{j-2}(\alpha, \beta)r_4 - \alpha\beta T_{j-3}(\alpha, \beta)r_2, \qquad (A.10)$$

$$r_{2j-1} = T_{j-2}(\alpha, \beta)r_3 - \alpha\beta T_{j-3}(\alpha, \beta)r_1, \qquad (A.11)$$

is a solution of (II.6,II.7,A.4). The boundary conditions determine,

$$
\begin{aligned}
r_1 &= V_L, \quad r_2 = \epsilon - \mathrm{i}u, \quad r_3 = W_R + V_L(\alpha + \beta), \\
r_4 &= (\epsilon - \mathrm{i}u)\left(2\frac{W_R}{V_L} + \alpha + \beta\right).
\end{aligned} \qquad (A.12)
$$

where $r_1 = V_L$ is an arbitrary choice. The conditions (A.8) give $\{\epsilon, \beta\}$ in terms of $\alpha$,

$$
\begin{aligned}
\epsilon &= \pm\sqrt{V_L V_R + W_L W_R + V_L W_L(\alpha + \beta) - u^2}, \\
\beta &= \frac{V_R W_R}{V_L W_L}\frac{1}{\alpha}.
\end{aligned} \qquad (A.13)
$$

Using the parametrization

$$\alpha = \frac{\sqrt{V_R}\sqrt{W_R}}{\sqrt{V_L}\sqrt{W_L}} \exp(\mathrm{i}\theta), \qquad (A.14)$$

and then

$$T_n\left(\alpha, \frac{V_R W_R}{V_L W_L}\frac{1}{\alpha}\right) = \frac{V_R^{\frac{n}{2}} W_R^{\frac{n}{2}}}{V_L^{\frac{n}{2}} W_L^{\frac{n}{2}}} \frac{\sin((n+1)\theta)}{\sin(\theta)} \qquad (A.15)$$

in (A.10,A.11) we find (IV.10,IV.11) with the initial conditions (A.12).

The boundary condition at the end of the chain gives a condition on the parameter $\theta$, which depends on whether $N$ is odd or even. For odd $N$, the condition $x_{L+1} = 0$ comes from (IV.10) with $2n = L + 1$ and gives the quantized solution (IV.8). For even $N$, the condition $x_{L+1} = 0$ comes from (IV.11) with $2n - 1 = L + 1$ and leads to the transcendental equation (IV.9). The solution (IV.12,IV.13) is obtained analogously.

## Appendix B: Discriminant of the characteristic polynomial

We recall that a polynomial has repeated roots if and only if its discriminant

$$\mathrm{disc}(P_N(z)) = \mathrm{disc}\left(\det_N (\mathcal{T}_N - z)\right) \qquad (B.1)$$

vanishes. The discriminant is given by the determinant of the Sylvester matrix [56], and it is in general an intricate expression of the parameters of the model. The

vanishing of the discriminant (B.1) is a necessary condition for the existence of an exceptional point. Of course, the discriminant vanishes when the quasienergies collide. Additionally, for the model considered in this paper, the eigenvectors always coalesce when the quasienergies collide, recall the expressions (III.8,III.9,IV.10). Thus, the collision of quasienergies is also sufficient for the existence of an exceptional point.

In the main text, the conditions for gap closing/exceptional points are determined from the simple characterization of the eigenvalues by (III.7) and (IV.4). Here these results are verified by explicitly computing the discriminant (B.1) using `Mathematica` for different boundary conditions.

### 1. Closed boundary conditions

Let us consider $\gamma = 1$ and $u = 0$. For a lattice with $N = 10$ sites the discriminant (B.1) is given by,

$$
\begin{aligned}
\text{disc}(P_{10}^{(\text{PBC})}(z)) = {} & 10^{10} \left( W_L^5 + V_R^5 \right) \left( V_L^5 + W_R^5 \right) \\
& \times \left( V_L^5 W_L^5 - V_R^5 W_R^5 \right)^8.
\end{aligned} \tag{B.2}
$$

The roots of the factors $\left( W_L^5 + V_R^5 \right) \left( V_L^5 + W_R^5 \right)$ are clearly given by (III.17). The extra factor comes from (III.20), which can be seen by writing (III.20) like,

$$
\frac{V_L W_L}{V_R W_R} = e^{i(k+k')} = e^{\frac{2}{5} i \pi (\ell + \ell')} \tag{B.3}
$$

with $\ell + \ell' = 1, \ldots, 5$. As mentioned in the main text, this last factor produce degenerate eigenvalues of $\mathcal{T}_N$ but associated with different eigenvectors, thus they are not exceptional points.

Now let us consider again $N = 10$, $\gamma = 1$, but with $u \neq 0$. In this case the discriminant is given by,

$$
\begin{aligned}
\text{disc}(P_{10}^{(\text{PBC})}(z)) = {} & 10^{10} \left( V_L^5 W_L^5 - V_R^5 W_R^5 \right)^8 \\
& \times \left( \left( W_L + V_R \right) \left( V_L + W_R \right) - u^2 \right) U(u)
\end{aligned} \tag{B.4}
$$

where

$$
\begin{aligned}
U(u) = {} & u^8 + u^6 \left( V_L W_L + V_R W_R - 4 V_L V_R - 4 W_L W_R \right) \\
& + u^4 \big( 6 V_L^2 V_R^2 - 3 V_L^2 V_R W_L + V_L^2 W_L^2 - 3 V_L V_R^2 W_R \\
& \quad + 9 V_L V_R W_L W_R - 3 V_L W_L^2 W_R - 3 V_L W_L^2 W_R \\
& \quad + V_R^2 W_R^2 - 3 V_R W_L W_R^2 + 6 W_L^2 W_R^2 \big) \\
& + u^2 \big( - 4 V_L^3 V_R^3 + 3 V_L^3 V_R^2 W_L - 2 V_L^3 V_R W_L^2 + V_L^3 W_L^3 \\
& \quad + 3 V_L^2 V_R^3 W_R - 6 V_L^2 V_R^2 W_L W_R + 4 V_L^2 V_R W_L^2 W_R \\
& \quad - 2 V_L^2 W_L^3 W_R - 2 V_L V_R^3 W_R^2 + 4 V_L V_R^2 W_L W_R^2 \\
& \quad - 6 V_L V_R W_L^2 W_R^2 + 3 V_L W_L^3 W_R^2 + V_R^3 W_R^3 \\
& \quad - 2 V_R^2 W_L W_R^3 + 3 V_R W_L^2 W_R^3 - 4 W_L^3 W_R^3 \big) \\
& + \left( - W_L V_R^3 + W_L^2 V_R^2 - W_L^3 V_R + W_L^4 + V_R^4 \right) \\
& \times \left( - V_L^3 W_R + V_L^2 W_R^2 - V_L W_R^3 + V_L^4 + W_R^4 \right).
\end{aligned} \tag{B.5}
$$

The factor $\left( V_L^5 W_L^5 - V_R^5 W_R^5 \right)$ is also present for the non-zero field case, and is solved by (B.3), however it does not lead to exceptional points. To analyze the factor $\left( \left( W_L + V_R \right) \left( V_L + W_R \right) - u^2 \right) U(u)$ we recall (III.16) for every possible quantized $k$ (five for $N = 10$). For example, for $k = 2\pi$,

$$
H_1(2\pi) H_2(2\pi) - u^2 = \left( W_L + V_R \right) \left( V_L + W_R \right) - u^2. \tag{B.6}
$$

Actually, we can check by brute force the following factorization,

$$
\begin{aligned}
& \left( \left( W_L + V_R \right) \left( V_L + W_R \right) - u^2 \right) U(u) = \\
& \prod_{m=1}^{5} H_1(k_m) H_2(k_m) - u^2
\end{aligned} \tag{B.7}
$$

and confirm that the exceptional points give the roots of the discriminant.

### 2. Open boundary conditions

Now let us consider $N = 11$, $\gamma = 0$ (OBC) and $u = 0$. In this case the discriminant (B.1) is given by,

$$
\text{disc}(P_{11}^{(\text{OBC})}(z)) = 48922361856 V_L^{20} V_R^{20} W_L^{20} W_R^{20} f_1^3 f_2^3 f_3^3 \tag{B.8}
$$

where

$$
\begin{aligned}
f_1 &= V_L V_R + W_L W_R \\
f_2 &= V_L^2 V_R^2 - V_L V_R W_L W_R + W_L^2 W_R^2 \\
f_3 &= V_L^2 V_R^2 + V_L V_R W_L W_R + W_L^2 W_R^2.
\end{aligned} \tag{B.9}
$$

Of course, the discriminant (B.8) vanishes when any of the couplings vanishes, consistent with our findings in the main text. The remaining factors can be verified by rewriting

$$
f_1 f_2 f_3 = V_L^5 V_R^5 \sum_{j=0}^{5} \left( \frac{1}{\sqrt{\delta}} \right)^{2j} \tag{B.10}
$$

whose ten roots are given by (IV.16).

Next, we turn on the field $u$. In this case, we have,

$$
\begin{aligned}
\text{disc}(P_{11}^{(\text{OBC})}(z)) = {} & -48922361856 V_L^{20} V_R^{20} W_L^{20} W_R^{20} \\
& \times f_1^2 f_2^2 f_3^2 g_1 g_2 g_3
\end{aligned} \tag{B.11}
$$

where $f_j$ is given by (B.9) and $g_j$ are given by

$$
\begin{aligned}
g_1 &= u^2 - \left( V_L V_R + W_L W_R \right), \\
g_2 &= u^4 - 2 u^2 f_1 + f_2, \\
g_3 &= u^4 - 2 u^2 f_1 + f_3.
\end{aligned} \tag{B.12}
$$

Part of the roots remain the same as in the zero field case, but field dependent factors $g_j$ appear. These are associated with (IV.20). Indeed, by brute force computation we can verify

$$
\prod_{m=1}^{5} u^2 - \tilde{H}_1(\theta_m) \tilde{H}_2(\theta_m) = g_1 g_2 g_3 \tag{B.13}
$$

where $\theta_m$ is given by (IV.8). Let us remark that the product $g_1 g_2 g_3$ only depends on $\bar{u}$ and $\sqrt{\delta}$, up to an overall factor.

Finally, let us consider $N = 12$ with non-zero field $u$. The discriminant is given by,

$$\text{disc}(P_{12}^{(\text{OBC})}(z)) = 4096 V_L^{20} V_R^{20} W_L^{30} W_R^{30} U_0 U_1(u) \tag{B.14}$$

where

$$
\begin{aligned}
U_0 &= 153664 V_L^5 V_R^5 + 98000 V_L^4 V_R^4 W_L W_R \\
&+ 60152 V_L^3 V_R^3 W_L^2 W_R^2 + 34709 V_L^2 V_R^2 W_L^3 W_R^3 \\
&+ 17856 V_L V_R W_L^4 W_R^4 + 6912 W_L^5 W_R^5 \tag{B.15}
\end{aligned}
$$

and

$$
\begin{aligned}
U_1(u) &= u^{12} - u^{10}(6 V_L V_R + 5 W_L W_R) \\
&+ 5u^8(3 V_L^2 V_R^2 + 4 V_L V_R W_L W_R + 2 W_L^2 W_R^2) \\
&- 2u^6(10 V_L^3 V_R^3 + 15 V_L^2 V_R^2 W_L W_R + 12 V_L V_R W_L^2 W_R^2 \\
&\quad + 5 W_L^3 W_R^3) \\
&+ u^4(15 V_L^4 V_R^4 + 20 V_L^3 V_R^3 W_L W_R + 18 V_L^2 V_R^2 W_L^2 W_R^2 \\
&\quad + 12 V_L V_R W_L^3 W_R^3 + 5 W_L^4 W_R^4) \\
&- u^2(6 V_L^5 V_R^5 + 5 V_L^4 V_R^4 W_L W_R + 4 V_L^3 V_R^3 W_L^2 W_R^2 \\
&\quad + 3 V_L^2 V_R^2 W_L^3 W_R^3 + 2 V_L V_R W_L^4 W_R^4 + W_L^5 W_R^5) \\
&+ V_L^6 V_R^6. \tag{B.16}
\end{aligned}
$$

Again, it is clear that the vanishing of any of the coupling parameters leads to a null discriminant (B.14). The factor $U_0$ can be written as

$$
\begin{aligned}
U_0 &= 153664 V_L^5 V_R^5 \Big( 1 + \frac{125}{196} \frac{1}{\sqrt{\delta}^2} + \frac{7519}{19208} \frac{1}{\sqrt{\delta}^4} \\
&+ \frac{34709}{153664} \frac{1}{\sqrt{\delta}^6} + \frac{279}{2401} \frac{1}{\sqrt{\delta}^8} + \frac{108}{2041} \frac{1}{\sqrt{\delta}^{10}} \Big). \tag{B.17}
\end{aligned}
$$

This factor is independent of the field $u$, and we can check numerically that the ten roots are given by (IV.25,IV.26), see summary in Table I. The field dependent factor $U_1(u)$ can also be written in terms of $\sqrt{\delta}$ and $\bar{u} = u/(\sqrt{W_L}\sqrt{W_R})$, namely,

$$
\begin{aligned}
U_1(u) &= V_L^6 V_R^6 \Bigg( 1 - \frac{\bar{u}(3\bar{u}+1)}{\sqrt{\delta}^2} + \frac{\bar{u}(\bar{u}+1)(3\bar{u}^2 - \bar{u} - 1)}{\sqrt{\delta}^4} \\
&- \frac{\bar{u}(\bar{u}-1)^2(\bar{u}+1)^3}{\sqrt{\delta}^6} \Bigg) \Bigg( 1 - \frac{\bar{u}(3\bar{u}-1)}{\sqrt{\delta}^2} \\
&+ \frac{\bar{u}(\bar{u}-1)(3\bar{u}^2 + \bar{u} - 1)}{\sqrt{\delta}^4} - \frac{\bar{u}(\bar{u}+1)^2(\bar{u}-1)^3}{\sqrt{\delta}^6} \Bigg). \tag{B.18}
\end{aligned}
$$

We verify numerically that (IV.22,IV.23) do provide the roots of $U_1(u) = 0$. In Table II we consider $\bar{u} = 1/2$ and the associated solutions of (IV.22), as well as the associated exceptional $\sqrt{\delta}$'s.

---

[1] G. Lindblad, On the generators of quantum dynamical semigroups, Communications in Mathematical Physics 48, 119 (1976).

[2] F. Roccati, G. M. Palma, F. Bagarello, and F. Ciccarello, Non-Hermitian Physics and Master Equations, Open Systems & Information Dynamics 29, 2250004 (2022), arXiv:2201.05367 [quant-ph].

[3] M.-A. Miri and A. Alù, Exceptional points in optics and photonics, Science 363, eaar7709 (2019).

[4] R. Su, E. Estrecho, D. Biegańska, Y. Huang, M. Wurdack, M. Pieczarka, A. G. Truscott, T. C. H. Liew, E. A. Ostrovskaya, and Q. Xiong, Direct measurement of a non-hermitian topological invariant in a hybrid light-matter system, Science Advances 7, eabj8905 (2021), https://www.science.org/doi/pdf/10.1126/sciadv.abj8905.

[5] Y. Yang, Y.-P. Wang, J. W. Rao, Y. S. Gui, B. M. Yao, W. Lu, and C.-M. Hu, Unconventional singularity in anti-parity-time symmetric cavity magnonics, Phys. Rev. Lett. 125, 147202 (2020).

[6] M. Naghiloo, A. M., Y. Joglekar, and K. Murch, Quantum state tomography across the exceptional point in a single dissipative qubit, Nat. Phys. 15, 1232 (2019).

[7] Y. Ashida, Z. Gong, and M. Ueda, Non-Hermitian physics, Adv. Phys. 69, 249 (2021), arXiv:2006.01837 [cond-mat.mes-hall].

[8] D. Bernard and A. LeClair, A classification of non-hermitian random matrices, in Statistical Field Theories (Springer Netherlands, 2002) p. 207–214.

TABLE I. The ten roots of Eq. (B.17).

| $\theta'_{\text{EP}}$ | $\sqrt{\delta}$ |
|---|---|
| $0.580505 - 0.209455i$ | $-0.628308 + 0.408561i$ |
| $0.580505 + 0.209455i$ | $-0.628308 - 0.408561i$ |
| $1.07762 - 0.24362i$ | $-0.344178 + 0.638086i$ |
| $1.07762 + 0.24362i$ | $-0.344178 - 0.638086i$ |
| $1.5708 - 0.252951i$ | $0.718376$ |
| $1.5708 + 0.252951i$ | $-0.718376$ |
| $2.06398 - 0.24362i$ | $0.344178 + 0.638086i$ |
| $2.06398 + 0.24362i$ | $0.344178 - 0.638086i$ |
| $2.56109 - 0.209455i$ | $0.628308 + 0.408561i$ |
| $2.56109 + 0.209455i$ | $0.628308 - 0.408561i$ |

[9] K. Esaki, M. Sato, K. Hasebe, and M. Kohmoto, Edge states and topological phases in non-Hermitian systems, Phys. Rev. B 84, 205128 (2011), arXiv:1107.2079 [cond-mat.mes-hall].

[10] T. E. Lee, Anomalous Edge State in a Non-Hermitian Lattice, Phys. Rev. Lett. 116, 133903 (2016), arXiv:1603.05312 [quant-ph].

[11] S. Lieu, Topological phases in the non-Hermitian Su-Schrieffer-Heeger model, Phys. Rev. B 97, 045106 (2018), arXiv:1709.03788 [cond-mat.mes-hall].

TABLE II. We consider $\bar{u} = 0.5$ First six lines, + solutions, last six lines − solutions.

| $\theta'_{\mathrm{EP}}$ | $\sqrt{\delta}$ |
|---|---|
| $0.642557 - 0.0955613\mathrm{i}$ | $-0.673088 + 0.294372\mathrm{i}$ |
| $0.642557 + 0.0955613\mathrm{i}$ | $-0.673088 - 0.294372\mathrm{i}$ |
| $1.5708 - 0.191123\mathrm{i}$ | $0.694828\mathrm{i}$ |
| $1.5708 + 0.191123\mathrm{i}$ | $-0.694828\mathrm{i}$ |
| $2.49904 - 0.0955613\mathrm{i}$ | $0.673088 + 0.294372\mathrm{i}$ |
| $2.49904 + 0.0955613\mathrm{i}$ | $0.673088 - 0.294372\mathrm{i}$ |
| $0.343297$ | $-1.31139$ |
| $1.10745 - 0.172812\mathrm{i}$ | $-0.36085 + 0.604218\mathrm{i}$ |
| $1.10745 + 0.172812\mathrm{i}$ | $-0.36085 - 0.604218\mathrm{i}$ |
| $2.03414 - 0.172812\mathrm{i}$ | $0.36085 + 0.604218\mathrm{i}$ |
| $2.03414 + 0.172812\mathrm{i}$ | $0.36085 - 0.604218\mathrm{i}$ |
| $2.7983$ | $1.31139$ |

[12] S. Yao and Z. Wang, Edge States and Topological Invariants of Non-Hermitian Systems, Phys. Rev. Lett. 121, 086803 (2018), arXiv:1803.01876 [cond-mat.mes-hall].

[13] C. Yin, H. Jiang, L. Li, R. Lü, and S. Chen, Geometrical meaning of winding number and its characterization of topological phases in one-dimensional chiral non-Hermitian systems, Phys. Rev. A 97, 052115 (2018), arXiv:1802.04169 [cond-mat.mes-hall].

[14] K. Kawabata, K. Shiozaki, M. Ueda, and M. Sato, Symmetry and Topology in Non-Hermitian Physics, Physical Review X 9, 041015 (2019), arXiv:1812.09133 [cond-mat.mes-hall].

[15] E. J. Bergholtz, J. C. Budich, and F. K. Kunst, Exceptional topology of non-Hermitian systems, Rev. Mod. Phys. 93, 015005 (2021), arXiv:1912.10048 [cond-mat.mes-hall].

[16] K. Ding, C. Fang, and G. Ma, Non-Hermitian topology and exceptional-point geometries, Nature Rev. Phys. 4, 745 (2022), arXiv:2204.11601 [quant-ph].

[17] A. Altland and M. R. Zirnbauer, Nonstandard symmetry classes in mesoscopic normal-superconducting hybrid structures, Physical Review B 55, 1142–1161 (1997).

[18] Z. Gong, Y. Ashida, K. Kawabata, K. Takasan, S. Higashikawa, and M. Ueda, Topological phases of non-hermitian systems, Phys. Rev. X 8, 031079 (2018), arXiv:1802.07964 [cond-mat.mes-hall].

[19] F. K. Kunst, E. Edvardsson, J. C. Budich, and E. J. Bergholtz, Biorthogonal bulk-boundary correspondence in non-hermitian systems, Physical review letters 121, 026808 (2018).

[20] W. D. Heiss, The physics of exceptional points, J. Phys. A 45, 444016 (2012), arXiv:1210.7536 [quant-ph].

[21] W. P. Su, J. R. Schrieffer, and A. J. Heeger, Solitons in polyacetylene, Phys. Rev. Lett. 42, 1698 (1979).

[22] C. Ling, M. Xiao, C. T. Chan, S. F. Yu, and K. H. Fung, Topological edge plasmon modes between diatomic chains of plasmonic nanoparticles, Optics express 23, 2021 (2015).

[23] S. Longhi, D. Gatti, and G. D. Valle, Robust light transport in non-Hermitian photonic lattices, Scientific Reports 5, 13376 (2015), arXiv:1503.08787 [physics.optics].

[24] N. Hatano and D. R. Nelson, Localization transitions in non-hermitian quantum mechanics, Physical Review Letters 77, 570 (1996).

[25] K. Monkman and J. Sirker, Hidden zero modes and topology of multiband non-hermitian systems (2024), arXiv:2405.09728.

[26] M. Fossati, F. Ares, and P. Calabrese, Symmetry-resolved entanglement in critical non-Hermitian systems, Phys. Rev. B 107, 205153 (2023), arXiv:2303.05232 [cond-mat.stat-mech].

[27] L. Herviou, N. Regnault, and J. H. Bardarson, Entanglement spectrum and symmetries in non-Hermitian fermionic non-interacting models, SciPost Physics 7, 069 (2019), arXiv:1908.09852 [cond-mat.mes-hall].

[28] P.-Y. Chang, J.-S. You, X. Wen, and S. Ryu, Entanglement spectrum and entropy in topological non-Hermitian systems and nonunitary conformal field theory, Phys. Rev. Res. 2, 033069 (2020), arXiv:1909.01346 [cond-mat.str-el].

[29] W. Chen, L. Peng, H. Lu, and X. Lu, Characterizing bulk-boundary correspondence of one-dimensional non-hermitian interacting systems by edge entanglement entropy, Physical Review B 105, 10.1103/physrevb.105.075126 (2022).

[30] S.-X. Hu, Y. Fu, and Y. Zhang, Nontrivial worldline winding in non-Hermitian quantum systems, Phys. Rev. B 108, 245114 (2023), arXiv:2307.01260 [quant-ph].

[31] M. Brunelli, C. C. Wanjura, and A. Nunnenkamp, Restoration of the non-Hermitian bulk-boundary correspondence via topological amplification, SciPost Phys. 15, 173 (2023), arXiv:2207.12427 [quant-ph].

[32] L. Herviou, J. H. Bardarson, and N. Regnault, Defining a bulk-edge correspondence for non-hermitian hamiltonians via singular-value decomposition, Physical Review A 99, 10.1103/physreva.99.052118 (2019), arXiv:1901.00010 [cond-mat.mes-hall].

[33] R. Arouca, C. H. Lee, and C. Morais Smith, Unconventional scaling at non-hermitian critical points, Physical Review B 102, 10.1103/physrevb.102.245145 (2020), arXiv:2009.03541 [cond-mat.stat-mech].

[34] D. Halder, S. Ganguly, and S. Basu, Properties of the non-Hermitian SSH model: role of symmetry, J. Phys. Condens. Matter 35, 105901 (2023), arXiv:2209.13838 [quant-ph].

[35] R. Aquino, N. Lopes, and D. G. Barci, Critical and noncritical non-Hermitian topological phase transitions in one-dimensional chains, Phys. Rev. B 107, 035424 (2023), arXiv:2208.14400 [cond-mat.str-el].

[36] T. Kitagawa, E. Berg, M. Rudner, and E. Demler, Topological characterization of periodically driven quantum systems, Phys. Rev. B 82, 235114 (2010), arXiv:1010.6126 [cond-mat.mes-hall].

[37] D. Leykam, K. Y. Bliokh, C. Huang, Y. D. Chong, and F. Nori, Edge Modes, Degeneracies, and Topological Numbers in Non-Hermitian Systems, Phys. Rev. Lett. 118, 040401 (2017), arXiv:1610.04029 [cond-mat.mes-hall].

[38] H. Shen, B. Zhen, and L. Fu, Topological band theory for non-hermitian hamiltonians, Phys. Rev. Lett. 120, 146402 (2018).

[39] J. C. Garrison and E. M. Wright, Complex geometrical phases for dissipative systems, Physics Letters A 128, 177 (1988).

[40] G. Dattoli, R. Mignani, and A. Torre, Geometrical phase in the cyclic evolution of non-hermitian systems, Journal

of Physics A: Mathematical and General **23**, 5795 (1990).

[41] A. Mostafazadeh, A new class of adiabatic cyclic states and geometric phases for non-Hermitian Hamiltonians, Physics Letters A **264**, 11 (1999), arXiv:quant-ph/9911003 [quant-ph].

[42] S.-D. Liang and G.-Y. Huang, Topological invariance and global berry phase in non-hermitian systems, Phys. Rev. A **87**, 012118 (2013).

[43] H. Jiang, C. Yang, and S. Chen, Topological invariants and phase diagrams for one-dimensional two-band non-Hermitian systems without chiral symmetry, Phys. Rev. A **98**, 052116 (2018), arXiv:1809.00850 [cond-mat.mes-hall].

[44] M. Gover, The eigenproblem of a tridiagonal 2-Toeplitz matrix, Linear Algebra and its Applications **197-198**, 63 (1994).

[45] B. C. Shin, A formula for eigenpairs of certain symmetric tridiagonal matrices, Bulletin of the Australian Mathematical Society **55**, 249–254 (1997).

[46] K. D. Ikramov, Shin's formulas for eigenpairs of symmetric tridiagonal 2-toeplitz matrices, Bulletin of the Australian Mathematical Society **59**, 119–120 (1999).

[47] J. Sirker, M. Maiti, N. P. Konstantinidis, and N. Sedlmayr, Boundary fidelity and entanglement in the symmetry protected topological phase of the SSH model, Journal of Statistical Mechanics: Theory and Experiment **2014**, P10032 (2014), arXiv:1406.7832 [cond-mat.str-el].

[48] E. Cheng, M. T. Batchelor, and D. Cocks, Topological analysis of the complex SSH model, arXiv e-prints , arXiv:2308.04626 (2023), arXiv:2308.04626 [cond-mat.str-el].

[49] P. Pfeuty, The one-dimensional Ising model with a transverse field, Annals of Physics **57**, 79 (1970).

[50] R. A. Henry and M. T. Batchelor, Exceptional points in the Baxter-Fendley free parafermion model, SciPost Phys. **15**, 016 (2023), arXiv:2301.11031 [cond-mat.stat-mech].

[51] A. Böttcher and B. Silbermann, *Introduction to large truncated Toeplitz matrices* (Springer (New York), 1999).

[52] I. Peschel, LETTER TO THE EDITOR: Calculation of reduced density matrices from correlation functions, Journal of Physics A Mathematical General **36**, L205 (2003), arXiv:cond-mat/0212631 [cond-mat].

[53] G. Vidal, J. I. Latorre, E. Rico, and A. Kitaev, Entanglement in Quantum Critical Phenomena, Phys. Rev. Lett. **90**, 227902 (2003), arXiv:quant-ph/0211074 [quant-ph].

[54] Y.-B. Guo, Y.-C. Yu, R.-Z. Huang, L.-P. Yang, R.-Z. Chi, H.-J. Liao, and T. Xiang, Entanglement entropy of non-Hermitian free fermions, J. Phys. Condens. Matter **33**, 475502 (2021), arXiv:2105.09793 [cond-mat.mes-hall].

[55] Y. Le Gal, X. Turkeshi, and M. Schirò, Volume-to-area law entanglement transition in a non-Hermitian free fermionic chain, SciPost Physics **14**, 138 (2023), arXiv:2210.11937 [cond-mat.stat-mech].

[56] I. M. Gelfand, M. M. Kapranov, and A. V. Zelevinsky, *Discriminants, Resultants, and Multidimensional Determinants* (Birkhäuser Boston, 1994).