# Peer review of "Bulk-Boundary Correspondence and Exceptional Points for a Dimerized Hatano-Nelson Model with Staggered Potentials"

_SciPost Physics_

## Round 1 · Referee Report · Anonymous (Referee 1) · 2024-12-1

Strengths

1- The manuscript discusses various parameter regimes of the dimerized Hatano-Nelson model.

2- It provides detailed analytical derivations for this model.

Weaknesses

1- The manuscript does not open a new path or research direction in the field of non-Hermitian systems.

2- A clear motivation/introduction for the paper's central goal is missing.

3- various regimes of the selected model has been extensively studied in the literature. The paper requires further references to support the present research.

4- The manuscript is long, and reorganizing certain parts may improve the readability of the current work.

Report

The authors explored the dimerized Hatano-Nelson model, focusing particularly on analytical derivations. Considering various boundary conditions, they investigated the arising phases of the model, presented the Bulk-Boundary correspondence~(BBC), and studied the entanglement entropy in various parameter regimes of their model. Here, the results presented are mainly analytically derived and numerically supported.

The manuscript is written to be accessible to a large audience with less expertise in the field. However, the novelty of the presented results does not seem to meet the criteria for SciPost Physics. After addressing the following points, I recommend being considered for SciPost Physics Core.

Requested changes

1- While the manuscript's title is Bulk-Boundary Correspondence and Exceptional Points, the paper also extensively discusses entanglement entropy. Although this discussion is well-written, it scatters the central goal of the paper suggested in the title/abstract. This suggests that varying the title to better represent the content of the manuscript might be an option.

2-For a non-Hermitian matrix $H$, the right and left eigenvectors satisfy $H |R \rangle = \epsilon |R \rangle $ and $H^{\dagger} |L \rangle = \epsilon^{*} |L \rangle $ with $\dagger$ imposing the conjugate transpose. However, in the manuscript for the (complex) hopping matrix, merely the transpose operation is considered. A clarification on this should be provided.

3- Summarizing various arising phases, their associated symmetries, and parameter regimes in a table or a subsection may improve the accessibility of different results in this work.

4-It is stated that "we can observe the merging of the zero
field phases $(v1,v2) = (0,1)$ and $(1,-1)$ into a single phase as
well as the bending of the phases boundaries." This observation merely occurs for phases close to $|V_{L}| \leq 1$ and $|W_{L}| \leq 1$. Is this related to the positive value of $u(=0.5)$? Could the authors provide a plot for $u=-0.5$? Is this because of the emergence of a real-line gap as $u$ is increased?

5-It would improve the readability of the manuscript if the authors considered transferring the paragraph on the PT-symmetric phases on page 13 (the first paragraph) to section III.

6- Here are some minor typos and comments:
* Appendix B is referred to before Appendix A.
* On page 7, below Eq.(IV.21) "by by" should be replaced with "by"
* On page 12, the second column, the fourth line from the bottom, "then" should be replaced with "than".

Recommendation

Accept in alternative Journal (see Report)

---

## Editorial Decision

resubmitted